# Self-supervised dynamic learning for long-term high-fidelity image transmission through unstabilized diffusive media

Ziwei Li [1,2,3,5] ✉, Wei Zhou[1,5], Zhanhong Zhou[1], Shuqi Zhang[1], Jianyang Shi [1,2], Chao Shen[1,2], Junwen Zhang[1,2], Nan Chi [1,2] ✉ & Qionghai Dai [1,4] ✉

Multimode fiber (MMF) which supports parallel transmission of spatially distributed information is a promising platform for remote imaging and capacity-enhanced optical communication. However, the variability of the scattering MMF channel poses a challenge for achieving long-term accurate transmission over long distances, of which static optical propagation modeling with calibrated transmission matrix or data-driven learning will inevitably degenerate. In this paper, we present a self-supervised dynamic learning approach that achieves long-term, high-fidelity transmission of arbitrary optical fields through unstabilized MMFs. Multiple networks carrying both long- and short-term memory of the propagation model variations are adaptively updated and ensembled to achieve robust image recovery. We demonstrate >99.9% accuracy in the transmission of 1024 spatial degree-of-freedom over 1 km length MMFs lasting over 1000 seconds. The long-term high-fidelity capability enables compressive encoded transfer of high-resolution video with orders of throughput enhancement, offering insights for artificial intelligence promoted diffusive spatial transmission in practical applications.

Propagation of optical fields for long distances is an essential requirement in remote imaging and optical communication applications. Optical fibers, especially the single-mode fiber, have been elaborately deployed in internet connection, however, they only allow for the transmission of a single Gaussian fundamental beam. Multimode fiber (MMF), on the other hand, permits hundreds to several thousand spatial modes to pass through. This makes them a promising platform for direct image transmission[1] and mode-division multiplexing transmission to boost the channel capacity[2,3], showing potential in applications such as compact endoscopy in bio-imaging[4–6], high-rate optical communication[2,7], and quantum key teleportation[8,9].

However, the inevitable spatial mode dispersion in the MMFs induces a complex optical field mixing of the input image as it propagates through the fiber, resulting in intense scrambling of the output intensity distribution. Image restoration through diffuse media entails precise characterization of the physical system. Methods have been developed that measure the MMF's forward process in the form of a complex transmission matrix (TM), and unscramble the output image via back-projection[10,11]. However, the calibration of TM with a huge number of elements is time-consuming and complicated in implementation[12]. Artificial intelligent approaches have been proposed to model the forward process or directly estimate the decoding process for image inference from the scrambled output[13–16]. Notwithstanding the notable progress in modeling the scattering process, it is foreseen that the instability of the MMF propagation induced by system drifting and environmental disturbance will accumulate and severely deviate from the characterized model after a long time, especially for long-distance transmission. As a consequence, most reported literature have experimented on short MMFs (<10 m)[17–19] and were impeded from practical long-term high-fidelity image

[1]School of Information Science and Technology, Fudan University, 200433 Shanghai, China. [2]Shanghai Engineering Research Center of Low-Earth-Orbit Satellite Communication and Applications, 200433 Shanghai, China. [3]Pujiang Laboratory, 200232 Shanghai, China. [4]Tsinghua University, 100084 Beijing, China. [5]These authors contributed equally: Ziwei Li, Wei Zhou. ✉e-mail: lizw@fudan.edu.cn; nanchi@fudan.edu.cn; qhdai@tsinghua.edu.cn

transmission. Therefore, developing a technique to support long-term accurate spatial information transmission over long distances for remote video transfer remains a challenge.

One method to tackle the variation of scattering is to train a general network using large sets of data describing diverse conditions[20,21]. However, this requires a long-duration training data collection and will hinder its flexible application in practice. Moreover, the generalized model often achieves poorer performance at a certain timepoint than using a 'specific' network trained on data acquired within small time durations. Another inspiring approach is to use the mixture-of-expert framework, in which multiple expert networks each responsible for one sub-problem are fused together to achieve better generalization[22]. This architecture has been successfully applied for image denoising[23,24] and phase retrieval[25] to improve reconstruction accuracy and robustness to noise. However, the implementations have so far been confined to static contexts.

In this work, we develop a dynamic-learning framework that can adaptively handle the time-varying optical propagation in long MMFs. The proposed network, termed multi-scale memory dynamic-learning network (MMDN), leverages the multi-expert framework to separately model the long- and short-term dynamics of the unstable MMF channel with multiple networks and combine the multi-scale memory by

adaptive weighted ensemble. The network parameters are dynamically updated over time in a self-supervised manner, by learning from currently predicted pseudo-labels to synthesize the optimal inverse transmission model for subsequent image inference. MMDN achieves adaptive and accurate tracing of the variations on the optical propagation model in MMF, enabling parallel transmission of 1024 spatial degrees of freedom through 1km-length fibers with >99.9% accuracy for over 1000s-duration. The high-fidelity performance enables efficient transmission of high-resolution video with several orders of throughput enhancement by using compressive encoding, showing the feasibility of long-term high-fidelity spatial transmission. The proposed dynamic memory framework opens up a new paradigm for demixing through unstabilized diffuse media.

## Results

### Multi-scale memory dynamic-learning network

We experimented on an intensity-modulated optical setup (see Fig. 1a and Supplementary Fig. 1). The input pattern coupled into the MMF is transformed into a complex distorted field at the distal end due to spatial mode dispersion. We calibrated and analyzed the variations of diverse 100 m and 1km-length MMF transmission channels and observed slowly changing system drifting (see Supplementary Fig. 2).

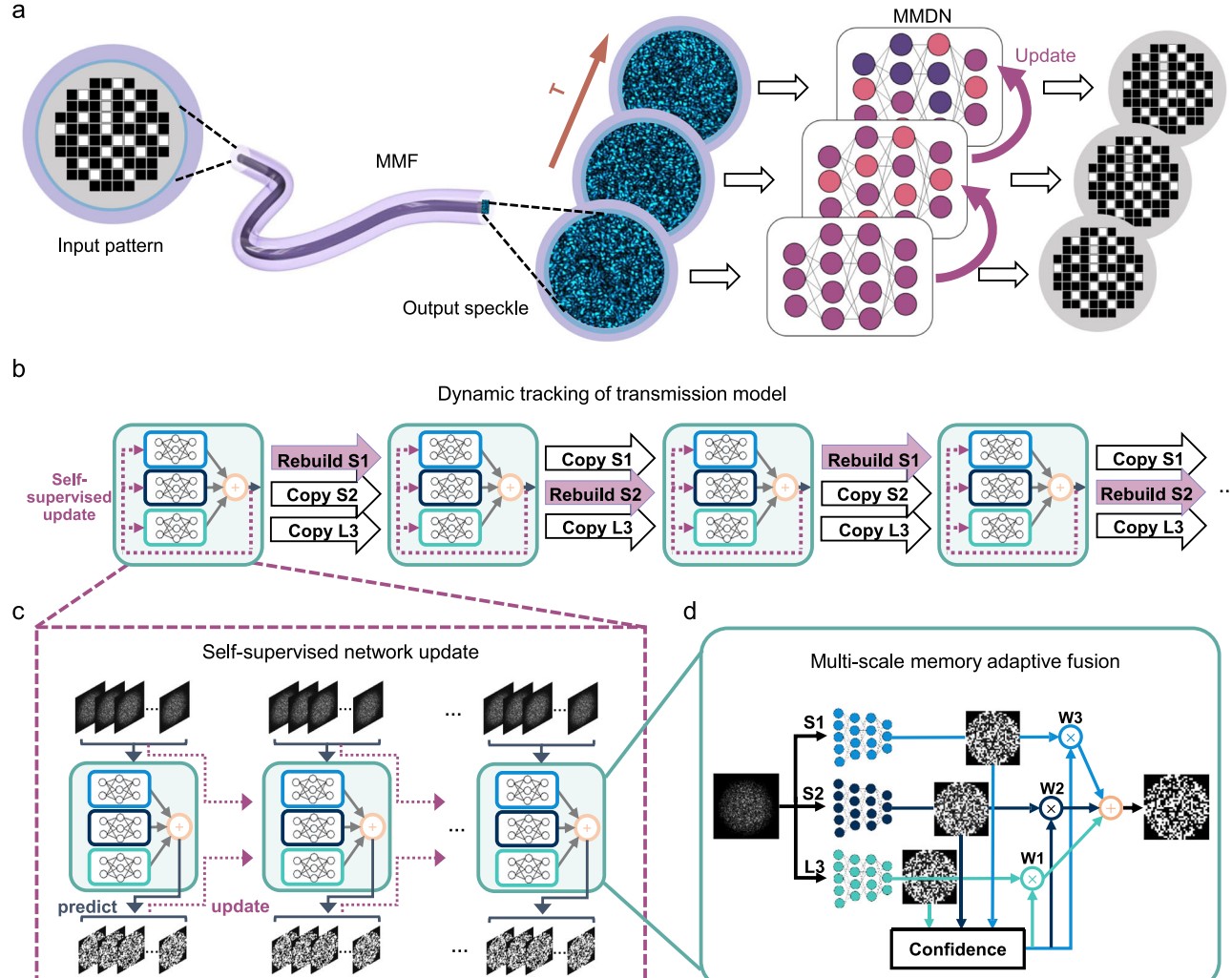

**Fig. 1 | Principle of dynamic learning for image transmission in varying diffuse media. a** Dynamic-learning framework for long-term data transmission in varying MMFs. **b, c** Workflow of dynamic subnetworks update for continuous data recovery, where **b** the short-term models are rebuilt interleavedly, and **c** each

submodel is frequently updated by self-supervised learning from previously predicted labels. **d** Multi-timescale expert networks with long- and short-term memories of the transmission characteristics ensembled adaptively by confidence-based weights.

Deep learning approaches have been reported to solve this nonlinear transformation problem to retrieve the input image from output speckle, however, one generalized neural network will fail to precisely model the gradually varying MMF channel over a long time. In this work, we proposed a dynamic-learning framework that adaptively tracks the optical propagation characteristics to achieve long-term image transmission with high accuracy through long MMFs. Instead of training a fixed and generalized neural network, we designed a compact network that generalizes well on short-term transmission. The MMDN model will be dynamically updated by online training on the unlabeled streaming data using predicted input images as labels, hence working in a self-supervised architecture (see Fig. 1c). According to the calibration results in Supplementary Fig. 2, the MMF transmission channels could sustain nearly unchanged within a 10-s duration, which ensures the feasibility of the self-supervised dynamic updating strategy.

Besides slow-varying system drift, we also witnessed intense jitter effects that will cause severe MMF channel deviations (illustrated in Supplementary Fig. 3). This physical prior knowledge inspires us to develop an ensemble framework of subnetworks which carry multi-scale memory of the system variations for the MMDN block. As shown in Fig. 1d, the output speckle image is fed into three networks, where S1 and S2 indicate two transmission models with different short-term memories, and L3 indicates one model with long-term memory. A grading module was additionally designed for confidence evaluation of expert subnetworks, allowing adaptive memory fusing to produce an optimized prediction that has good adaptability to both slow drifting and abrupt jitter effects. For continuous data transmission, the multi-scale models are self-supervised and updated at a short interval (i.e., 10 s). For S1 and S2 that aim to carry short-term information, we introduced the forgetting mechanism by alternatively wiping its memory of the accumulatively updated network model and rebuilding it with only the latest predicted data at specific intervals alternatively (see Methods); while the L3 will always keep the memory of previous transmission characterizations during the update process, as depicted in Fig. 1b. The proposed MMDN is supposed to show better performance in long-term data recovery through scattering MMFs, attributed to the enriched perspectives provided by the adaptively updated multi-timescale models.

### Long-term spatial transmission through unstabilized MMFs

We evaluated the performance of the proposed MMDN for long-term spatial transmission through diffuse media by experimenting on 100m- and 1km-length MMFs placed on the optical table without vibration isolation. To verify the ability to transmit arbitrary spatial information, we showed reconstruction of random binary patterns with spatial degree-of-freedom of $16 \times 16$ and $32 \times 32$, as opposed to the specified images such as the MNIST dataset (see Supplementary Video 1). We additionally reshaped the common grid pixel assignment to accommodate the round-shape property of the MMF entry, and proved slightly enhanced performance (see Supplementary Fig. 7). The output speckle patterns were captured by a camera at 100 fps. At the preparation stage of data transmission, a sequence of known patterns is transmitted, and the MMDN is statically trained on the initial instances. Once finished, the MMDN is ready for a dynamic update to support continuous image transmission.

In Fig. 2, we show the transmission results for over 1000 s on four different settings. The static neural network (StaticNN) trained on the first 500-s data and tested on the following 500-s is taken as the baseline method. The transmission matrix (TM) based reconstruction has also been investigated and compared (see Supplementary Fig. 6). The MMDN was dynamically updated at each 10-s interval. Comparing the reconstructed accuracy averaged on each 10-s batch, we observed an obvious degradation after 200 s using StaticNN, especially for 1 km

fibers; while MMDN achieved ~100% accuracy throughout the 1000 s period. Although the increase in fiber distance makes its scattering channel vary more intensely, the dynamic-learning ability of the proposed MMDN allows it to precisely model the dynamics and adapt to the current state of the physical process. We also illustrated representative recovered spatial profiles and the averaged spatial distribution of transmission accuracy. For 1 km transmission, MMDN enables <0.1% error rate across the spatial position and an average accuracy of 99.97% and 99.99% for $32 \times 32$-pixel and $16 \times 16$-pixel reconstruction, which is two orders improved than using StaticNN. For 100 m transmission cases, MMDN even reduces the average error rate to 2e-6 throughout the transmission period. As a result, we demonstrated that MMDN can permit long-term spatial information transmission with high fidelity.

Further, we tested on recovery of high-level encoded random images. The grayscale input patterns are generated by the binary DMD using spatial division multiplexing. Experimenting on the 1km-length MMF, we demonstrated long-term transmission of 2-bit images with $24 \times 24$-pixel (see Fig. 3a) and 4-bit images with $16 \times 16$-pixel (see Fig. 3b). The mean absolute errors (MAEs) of recovered non-binary patterns are measured in temporal and spatial domains. The accuracy metric (1-MAE) is above 0.994 for the 2-bit reconstruction and 0.992 for the 4-bit reconstruction during the 500-s period, and the average accuracy is 0.998 and 0.999, respectively. As a comparison, the StaticNN will intensely degrade in reconstruction accuracy as the system variation aggregates over time. The experiments proved the potential of the proposed MMDN approach for practical applications involving gray-level image delivery.

### Multi-scale memory ensemble improves instability robustness

The dynamic-learning strategy for MMF dynamics modeling intuitively raises a critical demand for accurate inference on the current batch, so as to provide reliable training sets for the subsequent update. However, two main factors exist that could hinder its naïve implementation from practical usage. First of all, the inevitable inference error for the previous batch induces a bias to the model update, which will accumulate and largely deteriorate the network effectiveness after a long period[26]. Besides, the abrupt disturbances happen frequently for unstabilized long MMFs, resulting in significant transmission characteristics distinguishment even between adjacent batches. MMDN incorporates the physical prior inspired multi-scale memory mixture architecture, providing a solution for precise and robust modeling of system dynamics.

We next investigated the benefits of long and short-term memory ensemble network design. We chose two variable MMF channels with medium instability including a 100m-length gradient-indexed MMF of 200 μm diameter (GI-200) and a 1km-length step-indexed MMF of 200 μm diameter (SI-200) for spatial transmission of $16 \times 16$ and $32 \times 32$-pixel random binary patterns. We analyzed the contributions of different subnetworks in MMDN by evaluating their intermediate prediction accuracy. For the GI-200 case, as shown in Fig. 4a, we presented the average accuracy over a 1000 s duration using S1, S2, L3, and the confidence-based ensemble network and observed that the multi-scale model improved the accuracy by around 10-fold compared to the short-term models and the long-term model. Comparing the time-varying transmission error rate shown in Fig. 3b, the long-term model shows degraded performance especially at the 40−55 s period, while the short-term models provide 'wiser' predictions and contribute to producing a good ensemble model. Similar comparisons in averaged transmission accuracy and error rate at each timestep are also presented for the SI-200 case (see Fig. 4c, d). In this experiment, the long-term model works better than short-term models most of the time, as the magnified time clip is within 40−45 s, and the multi-scale model outperforms all other single-scale models throughout the period. The above results verified that using a mixture of expert networks

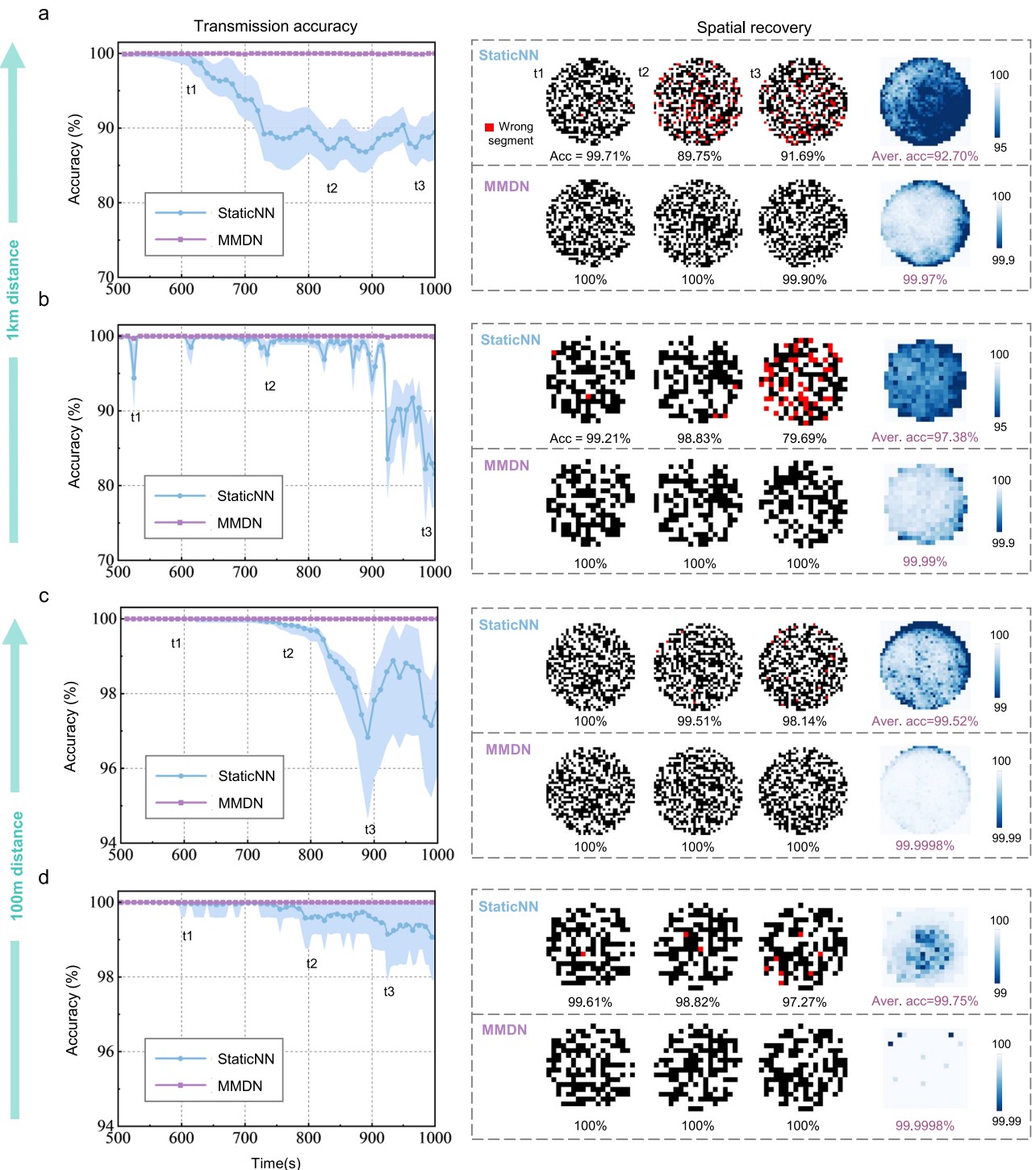

**Fig. 2 | Performance of spatial transmission over a long time in unstabilized MMFs.** Averaged transmission accuracy over time and accuracy spatial distribution of MMDN and static network are compared on four configurations with decreasing instability: **a** spatial resolution: 32 × 32, fiber length: 1 km; **b** spatial resolution: 16 × 16, fiber length: 1 km; **c** spatial resolution: 32 × 32, fiber length: 100 m; **d** spatial resolution: 16 × 16, fiber length: 100 m.

with different memory scales will enrich perspectives of the dynamic scattering channel by learning from the same data, and overcome diverse variations in MMFs to achieve spatial transmission with high accuracy over long periods.

### Scalability and generalization of MMDN

Having validated long-term high-fidelity transmission of generalized patterns, we also noted that the MMDN is well applicable to transmit specific types of natural images. In Fig. 5a, b, we show a set of recovered hand-written digits and fashion symbols transmitted through a 1km-long fiber. The input images of 28 × 28 pixels are upsampled to be 32 × 32 and binarized by the greyscale threshold of 0.5. The initial MMDN model trained on 20,000 paired data is then dynamically updated to support continuous image transmission of over 1000 s. To evaluate the quality of reconstructed images, we quantitively compared the SSIM and accuracy obtained using StaticNN and proposed

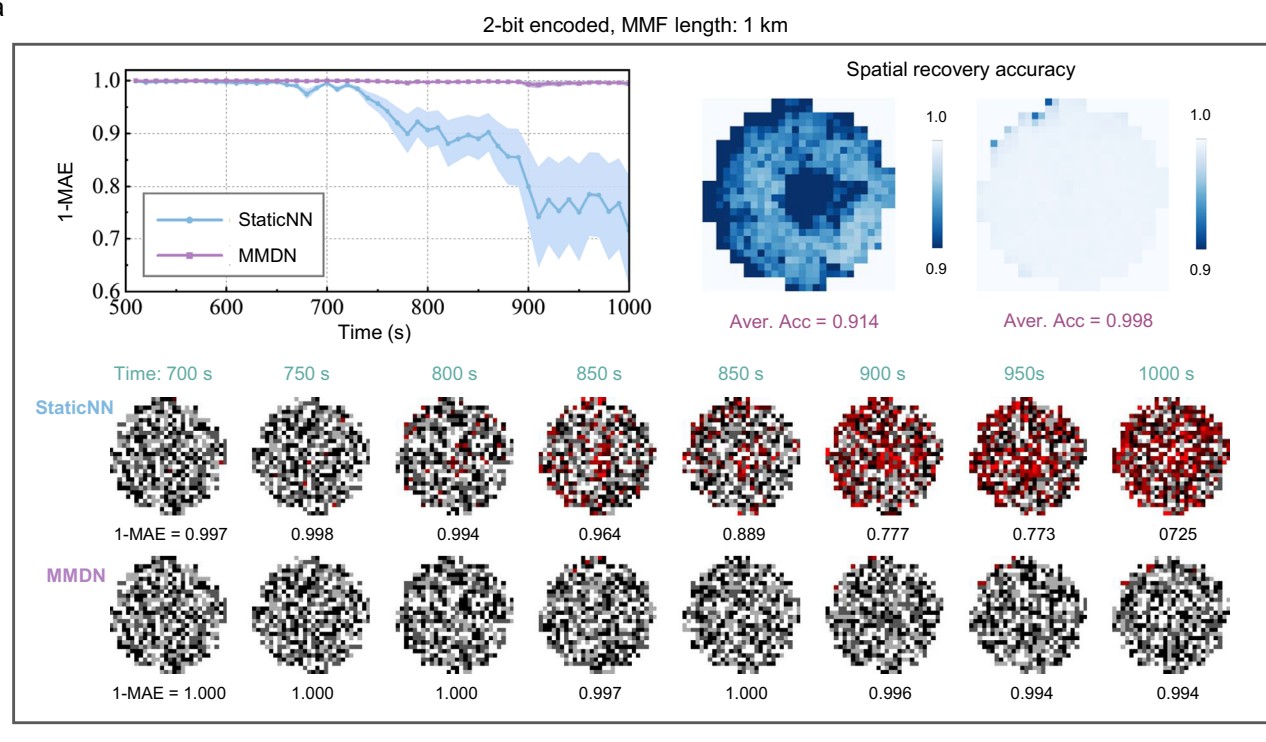

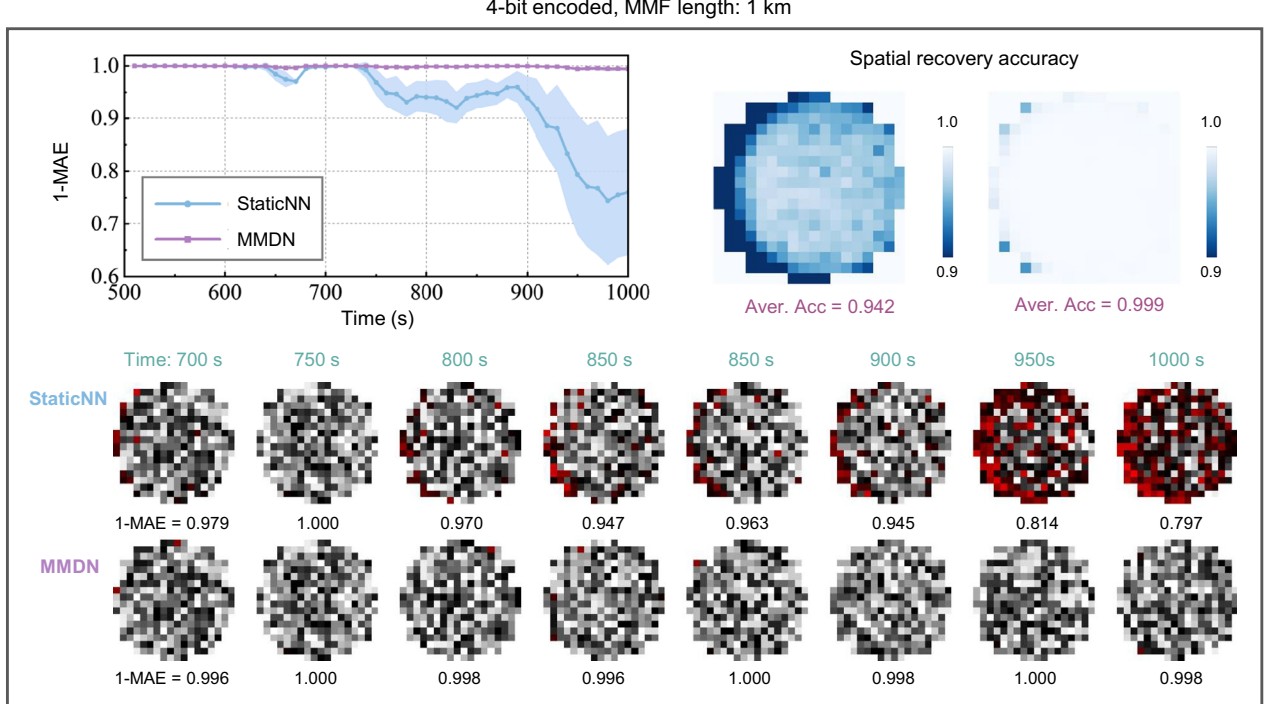

**Fig. 3 | Performance of non-binary encoded image transmission over a long time in unstablized MMF.** Averaged transmission accuracy over time and accuracy spatial distribution of MMDN and StaticNN on two non-binary encoded pattern transmissions: **a** 2-bit encoded patterns of 24 × 24-pixel resolution and **b** 4-bit encoded patterns of 16 × 16-pixel resolution. The MMF (Step-index, 200 µm core diameter, NA 0.22) is 1 km long.

MMDN. We observed nearly errorless transmission of both MNIST and fashion-MNIST datasets with MMDN during the whole period, and the SSIM is mostly above 0.99. Whereas, the static model will degrade in transmission accuracy and image similarity (see Supplementary Video 2). This indicates the feasibility of image transmission in unstablized scattering MMF using MMDN, laying the foundation for direct natural image transmission for remote imaging applications such as biological endoscopy.

One big concern of the learning-based image reconstruction approach is whether the network is able to transfer to other unseen categories of images. To demonstrate the generalized performance of the proposed network, we used the MMDN dynamically trained on Latin alphabet images for inference on the digit dataset (see Fig. 5c) and measured the transmission accuracy (see Fig. 5d). Examples of reconstructed input patterns right after the switch of image category as well as 10 s and 150 s after the switch are presented in Fig. 5e,

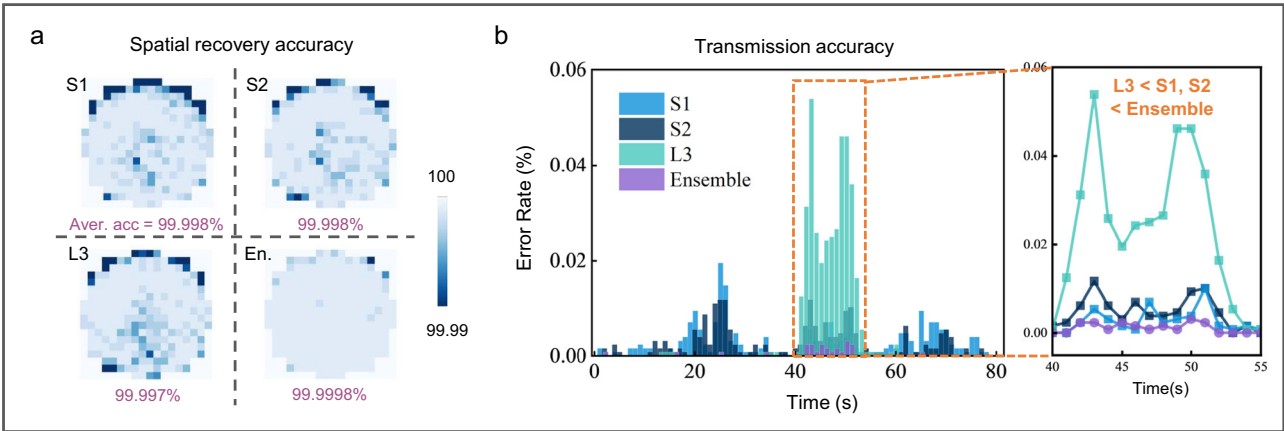

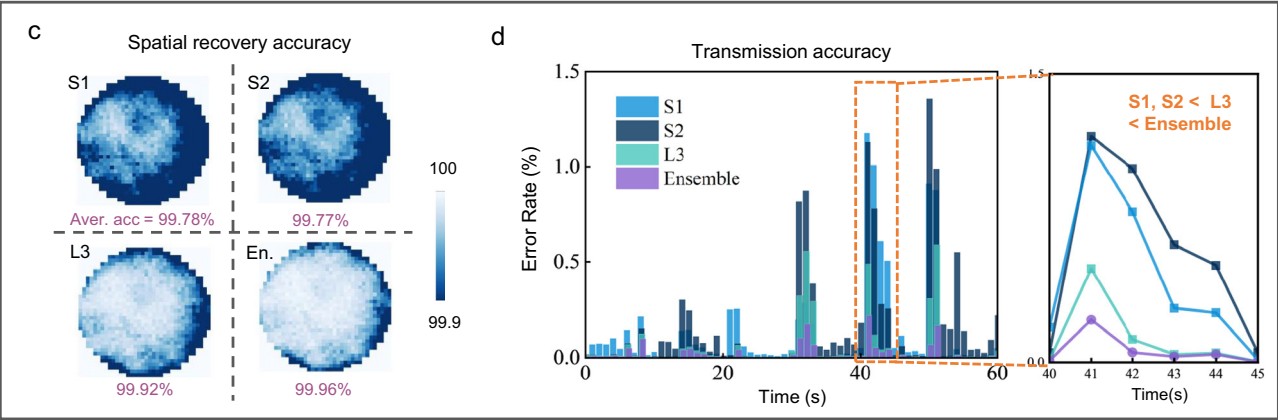

**Fig. 4 | Performance of multi-scale ensemble in MMDN.** Evaluation on averaged accuracy spatial distribution and error rate over time of **a**, **b** 16 × 16 spatial information transmission through 100 m GI MMF, and **c**, **d** 32 × 32 spatial information transmission through 1 km SI MMF, using S1, S2, L3, and the ensemble model.

respectively. We observed an average accuracy of 99.44% and good visual fidelity with a high SSIM of 0.978 even without fine-tuning the network (i.e., $T = 0$). The accuracy was further elevated up to 99.68% and the SSIM up to 0.989 after 2 batches (i.e., $T = 10$ s) attributed to the dynamic update ability of MMDN, and the high accuracy could be well retained during the following 150 s. This experiment demonstrates that the proposed MMDN has good generalization performance for image transmission.

### Long-term high-throughput video transmission

We further propose a practical high-throughput video transmission protocol via unstabilized long MMFs, as shown in Fig. 6. High-resolution video streams are compressively encoded into bit streams and spatially arranged to form a sequence of spatial-multiplexing-coded patterns. The spatial-multiplexed MMF setup can be used to support up to thousands of independent communication channels transmitting in parallel, and the proposed MMDN addresses the encoded spatial pattern recovery from recorded speckle images with nearly 100% accuracy to guarantee high-quality video decoding.

We experimented on the 1 km transmission of a 6-min full-color video of 480 × 480-pixel resolution and 20 fps. The video was encoded in the form of H.264 with a compression ratio of 1.39‰ and then converted into 32 × 32-pixel spatially multiplexed patterns. The proof-of-concept MMF setup with a 150 fps acquisition rate guarantees adequate bandwidth for real-time transmission of encoded video (see Supplementary Video 3). We evaluated the transmission accuracy of the spatial-multiplexed encoded signals in Fig. 6b and the

visual fidelity of decoded video frames using the SSIM metric in Fig. 6c. When using the static network for image recovery, we observed that a slight signal accuracy deterioration of less than 2% starting at 100 s would cause a significant drop in decoded video quality, leading to the SSIM of recovered video frames drastically dropping below 0.4. Noticeably, even when the transmission performance can be restored to good accuracy afterward, such as the 160-180 s period as highlighted with the asterisk in Fig. 6b, the video decoding corruption still remains irreparable. By comparison, our proposed MMDN ensures high-accuracy spatial transmission over the long term and hence can be incorporated into the highly compressed encoded video transmission protocol to achieve high-quality image recovery with the SSIMs all above 0.995 within the 6-min duration. Example recovered frames within the 93–99 s period are illustrated in Fig. 6d to visualize the decoding corruption caused by the slight transmission error using the static network. The results validated the superiority in decoding accuracy using MMDN for encoded video transmission, which offered over 700-fold efficiency enhancement compared to the plain coded video transmission scheme (see Supplementary Fig. 8).

### Discussion

Spatial reconstruction from diffusive MMF transmission is a challenging task and has so far been limited to short-distance fibers and faced the tradeoff between high accuracy and long-time generalization. In the work, we described the MMDN, a multi-scale dynamic-learning approach that can achieve >99.9% accuracy of long-time

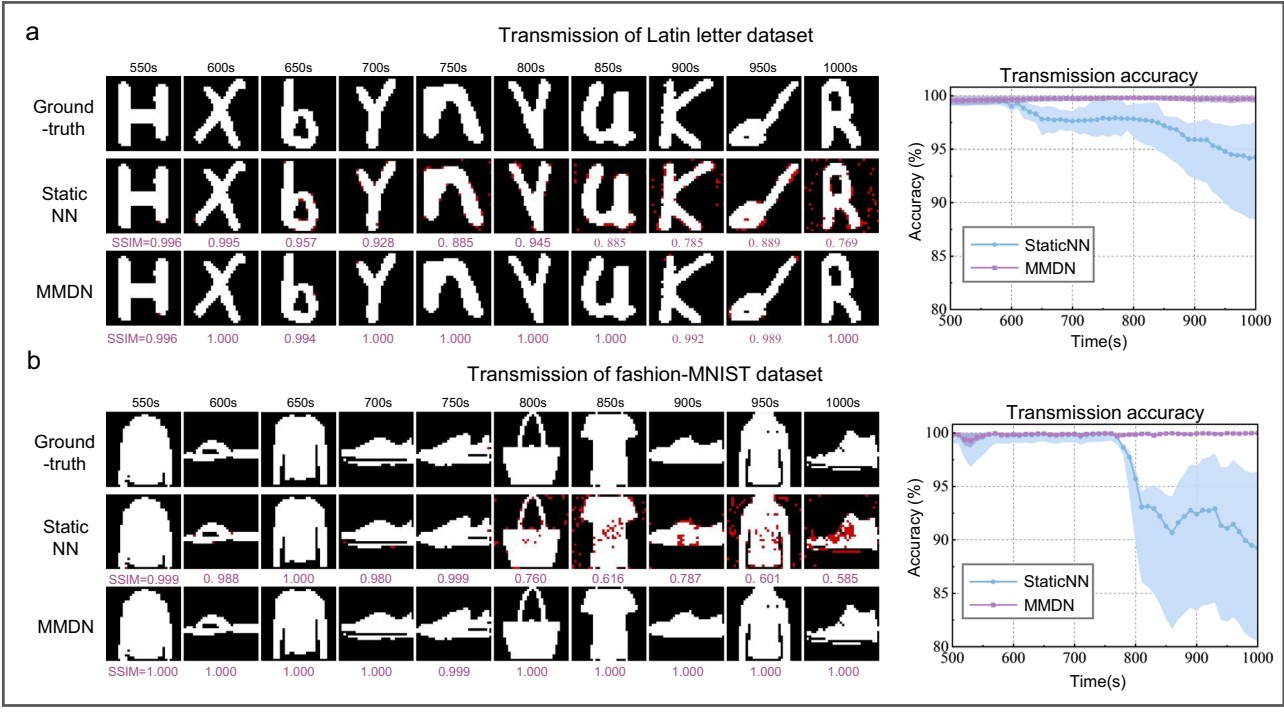

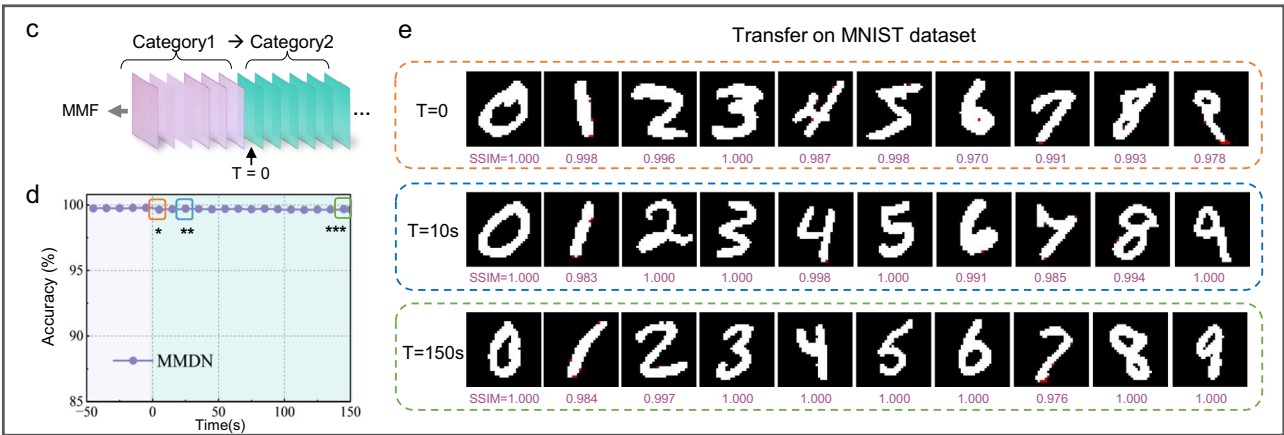

**Fig. 5 | Scalability to natural image transmission and generalization evaluation.** **a** Example E-MNIST images reconstructed with StaticNN and MMDN over 1000 s through 1 km fiber, and reconstruction accuracy over time and space domain. **b** Example of reconstructed fashion-MNIST images and accuracy performance. **c** Illustration of changing image category during transmission. **d** Transfer learning performance of reconstructing digits using MMDN trained on Latin alphabet letters dataset. **e** Example recovered images at T = 0 s, 10 s, 150 s after image category change.

image transmission through unstabilized 1 km MMF channels. The MMDN technique is developed based on the physically informed model of the diffusive MMF system that fully describes the long-term drifting and short-term abruption effects, therefore enabling good accommodation with the dynamics of the system characteristics. We compared MMDN with previous approaches in terms of time duration, reconstruction quality, transmission distance, etc., and demonstrated a comprehensive improvement of our approach (see Supplementary Fig. 9). In particular, the high-fidelity long-term transmission performance of MMDN will allow for efficient video transmission in a highly compressed scheme. The superior decoding capability will also facilitate practical applications such as minimally invasive endoscopic imaging, spatial-multiplexed optical communication, and optical security taking a diffusive channel as the decoding medium.

In contrast to conventional methods relying on the calibration of transmission matrices, deep learning provides an alternative pathway for addressing intricate reconstruction challenges, offering notable benefits such as enhanced accuracy and the avoidance of strict adherence to mathematical model priors. Nevertheless, the efficacy of deep learning is contingent upon a substantial collection of labeled samples for robust and generalized network training. We empirically examine the required size of the training dataset for MMDN, as illustrated in Supplementary Fig. 11.

One intuitive premise of the proposed dynamic-learning approach is that the scattering MMF channels drift at a relatively slow pace. Based on this assumption, the network model optimally trained on the previously recorded dataset will exhibit only minor deviations from being 'perfect' for the current input data. Therefore, the previous model acting as the pre-trained model can provide a good initialization of network parameters and facilitate the convergence of the network update. Another concern of the current proof-of-concept spatial transmission system is the limited spatial modulation speed of SLM which would limit the bandwidth of MMFs compared with that of SMFs. One promising and practical solution to break this limit is to use programmable lighting arrays as the spatial

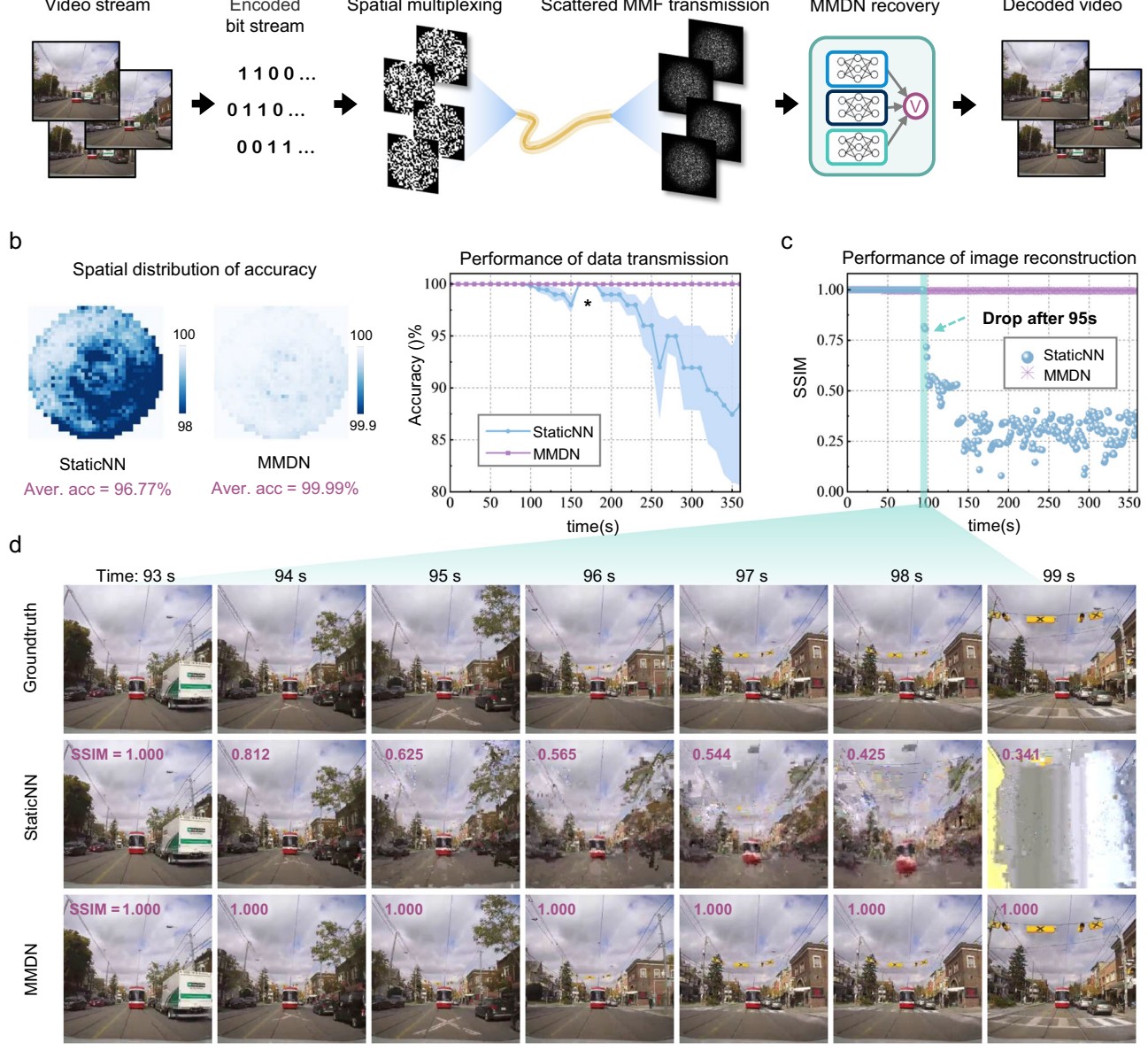

**Fig. 6 | Long-term transmission of compressive encoded high-throughput video. a** Pipeline of compressive encoded video transmission through spatial-multiplexed MMF channel and decoding with MMDN. **b** Transmission accuracy of encoded signals. **c** SSIM of decoded video frames. **d** Example video frames recovered with StaticNN and MMDN.

information encoder. Independently addressable LED arrays of $10 \times 10$ pixels with 100 MHz switching speed[27] and $64 \times 64$ pixels with 2.5 MHz[28] have been reported to generate fast-switching structured illumination patterns at low cost. Using a laser diode array, for example, the VCSEL[7] array, which is commonly used in LiDAR and data centers, spatial modulation with up to Gigahertz bandwidth can be achieved. Equipping with those high-speed spatial modulation modules, the bandwidth of MMFs could be elevated to hundreds of Gigahertz. On the other hand, the MMDN-enhanced MMF-based optical transmission technique gives rise to high-speed direct imaging that can achieve tens of kilohertz to megahertz frame rate (see Supplementary Fig. 12), extensively outperforms the SMF-based or wavefront-shaping enabled MMF-based raster-scanning remote imaging techniques.

## Methods

### Experimental setup

The optical setup for image transmission through the MMFs is described in Supplementary Fig. 1. The continuous-wave laser beam at wavelength 561 nm is expanded by a pair of lenses ($f_1 = 10$ mm, $f_2 = 100$ mm) and projected onto the SLM (V-7001) for binary amplitude modulation. The spatial modulated optical field is relayed by a pair of lenses ($f_3 = f_4 = 50$ mm) to the back-pupil plane of an objective lens (Nikon, 20X, 0.25NA) and coupled into the input facet of the MMF placed at the image plane of the objective. At the output of the MMF, the speckle field at the distal end is magnified by another objective (Nikon, 20X, 0.25NA) and imaged by the CMOS (MER2-230-167U3M). The specifications of the MMFs used in all experiments are summarized in Supplementary Fig. 10.

## Stability analysis of MMF channel

We adopted three metrics to evaluate the transmission channel stability over a large variety of MMFs, as summarized in Supplementary Fig. 2. The first metric measures the SSIM of a sequence of output speckle patterns to the reference pattern captured at $T = 0$ while transmitting a fixed image. The SSIM calculation can be expressed as

$$SSIM = \frac{\left(2 * \mu_x * \mu_y + C_1\right) * \left(2 * \sigma_{xy} + C_2\right)}{\left(\mu_x^2 + \mu_y^2 + C_1\right) * \left(\sigma_x^2 + \sigma_y^2 + C_2\right)}. \quad (1)$$

Here, $x$ and $y$ indicate the two images being compared, $\mu_x$, $\mu_y$ and $\sigma_x^2$, $\sigma_y^2$ are the mean values and variations of images $x$ and $y$, and $\sigma_{xy}$ is the covariance between two images. The $C_1$ and $C_2$ are two constants used to prevent the denominator from becoming zero or the result from becoming infinite.

The second metric measures the Helinger distance of the speckle energy distribution of two batches. The speckle energy distribution of each batch is obtained by summing up all speckle images in that batch, and one batch consists of 500 output speckle images captured within 5 s. For two discrete energy distributions $\mathbf{P} = (p_1, p_2 \ldots, p_n)$ and $\mathbf{Q} = (q_1, q_2 \ldots, q_n)$, $n$ is the total number of pixels in a single speckle image, the Hellinger distance can be calculated as

$$H^2(\mathbf{P}, \mathbf{Q}) = \frac{1}{2} \sum_{i=0}^{n} \left(\sqrt{q_i} - \sqrt{p_i}\right)^2. \quad (2)$$

The third metric computes the pixel-wise accuracy of the recovered image at each timepoint, by using an approximate reconstruction algorithm termed real-value inverse transmission matrix (RVITM)[29]. The test image is transmitted right after the approximal transmission matrix is calibrated. The accuracy for each recovered image is the percentage of mis-distinguished pixels to the full number of pixels.

## Data preparation

Our research employed arbitrary random binary images, as well as natural scene images including the Fashion-MNIST dataset, Latin letters from the E-MNIST dataset, and hand-written digitals from the MNIST dataset. The input images tested in the experiments are of $16 \times 16$-pixel and $32 \times 32$-pixel resolution. According to the maximum number of spatial modes that are allowed to transmit through the MMF, it is theoretically possible to input images of larger pixel resolution. The input images are reshaped in a round assignment as depicted in Supplementary Fig. 7. We used a binary SLM as the amplitude modulator to encode the spatial information, and the output speckles are captured by the monochromatic CMOS. To reduce computational complexity, we down-sampled the speckle images to $100 \times 100$-pixel and $150 \times 150$-pixel for reconstructing input patterns of $16 \times 16$-pixel and $32 \times 32$-pixel, respectively.

## Neural network architecture and training details

We designed an ensemble network consisting of three subnetworks with the same structure but updated differently over time. Each subnetwork is a convolutional neural network consisting of two convolutional layers and one fully-connected layer. Each convolution layer is followed by Dropout, Batch normalization, and ReLu activation. A sigmoid activation function is applied to the output layer for binary image reconstruction. We employed cross-entropy as the loss function and calculated the deviations from predicted images of each subnetwork to the digitized ensemble predictions. We trained the subnetworks by propagating the loss backward with the AdaDelta optimizer[30] at a learning rate of 0.1.

For the pretraining of MMDN, we use 5000 paired data for $16 \times 16$-pixel image reconstruction and 20000 paired data for $32 \times 32$-pixel image reconstruction. For each update, we use 500 paired data for $16 \times 16$-pixel image reconstruction and 1000 paired data for $32 \times 32$-pixel image reconstruction. Please refer to Supplementary Fig. 11 for a detailed comparison of MMDN performance training on different amounts of datasets. The training time for each round of network update processed on the workstation equipped with NVidia RTX3090 GPU is around 0.11 s and 0.98 s, respectively. We utilized early stopping to prevent overfitting and reduce the training time for each network update, and the network fine-tuning typically can be finished within 20 epochs.

All three modules S1, S2, and L3 are frequently fine-tuned on the newly input sequence at a 10-s interval for all experiments via the proposed self-supervised scheme. In addition, S1 and S2 modules are rebuilt alternatively at every five intervals to refresh their memory of the input data sequence and only keep short-term memory.

## Confidence-based ensemble algorithm

Each subnetwork of the ensemble network calculates the average confidence level of one predicted instance as:

$$c_k = \frac{1}{N} \sum_{i=1}^{N} \left(\left|P_k^i - \frac{1}{2}\right| * 1.8 + 0.1\right), \quad (3)$$

where $P_k^i$ represents the predicted value of pixel $i$ from the subnetwork $k$, which is binarized to be either 0 or 1, and $N$ is the spatial resolution of the input patterns. The confidence-based weight of each subnetwork is defined as:

$$w_k = \exp\left(\frac{\sum_{j \neq k} 1 - c_j}{\sum_j 1 - c_j}\right), j = 1,2,3. \quad (4)$$

The final ensemble prediction is the weighted summation of predictions $\mathbf{P}_1$, $\mathbf{P}_2$ and $\mathbf{P}_3$ from three subnetworks:

$$\mathbf{P} = \frac{w_1 * \mathbf{P}_1 + w_2 * \mathbf{P}_2 + w_3 * \mathbf{P}_3}{w_1 + w_2 + w_3}. \quad (5)$$

## Reporting summary

Further information on research design is available in the Nature Portfolio Reporting Summary linked to this article.

## Data availability

Source data are provided in this paper. One set of testing datasets to demonstrate the algorithm is provided in Figshare with the identifier https://doi.org/10.6084/m9.figshare.24249676. Source data are provided in this paper.

## Code availability

The code is available on GitHub (https://github.com/fudanawei/MMDN[31]).

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

## Acknowledgements

This work was sponsored by the National Natural Science Foundation of China (62231018, Z.L.; 61925104, N.C.), and Shanghai Science and Technology funding (2021SHZDZX0103, Z.L.).

## Author contributions

Z.L. conceived of the project; W.Z. developed and implemented the reconstruction algorithm; Z.L., S.Z., and W.Z. built the optical setup; Z.L., W.Z., and Z.Z. designed the experiments and analyzed the experimental data; W.Z., S.Z., and Z.Z. collected the data; J.S., C.S., and J.Z. provided instructions on the manuscript; N.C. and Q.D. provided mentoring support; Z.L. and W.Z. wrote the manuscript with input from all authors.

## Competing interests

The authors declare no competing interests.
