## [Peer Review File · Nature Communications]

REVIEWER COMMENTS

Reviewer #1 (Remarks to the Author):

In this manuscript, Zhou et al. report a dynamic neural network for image transmission through diffusive MMFs. The merits of the reported method include the claimed 99.9% reconstruction accuracy, self-supervised learning with no need for labelled data and the generalization to different image classes. However, the manuscript in its current form failed to explain the key points of their method, and the experimental setup might be too ideal to validate the performance of the reported network in real world applications.

Major points:

1. The most important part of the reported neural network, the self-supervised training, lacks essential details or explanations.

(a) It is not clear how the loss function is explicitly defined, and what regularization the authors used in network training.

(b) The convergence of the reported self-supervised training framework is questionable. The authors proposed to regard the previous network output as the label for current input data. But for the reported network architecture frequently updating by the input sequences, one can easily figure a trivial solution, in which the network produces constant for any input. The proof of correct convergence for the reported self-supervised training framework must be provided.

2. As a dynamic neural network, the update mechanism was hardly explained in the manuscript. How does the proposed CNN update its memory by its input sequence? What is the difference between S1, S2, and L3? What is the updating frequency for these three modules?

3. The experimental setup seems too ideal for validation of real-world applications. The inputs and the targets of the network were both binary images, which greatly reduces the difficulty of this reconstruction task and hinders the potent of the reported method for practical applications. The authors should at least show results on non-binary images, e.g., 2-bit, 4-bit images.

4. In addition to the static neural network baseline, I also wonder how conventional algorithms would perform against the reported method. Benchmarking would be good to include.

Minor points:

1. The specs of the MMF used in the experiment are not listed.

2. In line 219, please give a reference to AdaDelta paper.

3. The number of trainable parameters, and the training time on the given machine are not mentioned in the Methods section.

4. For Fig. 1b, the pink texts on the left, what is “self-s update”?

Reviewer #2 (Remarks to the Author):

The authors demonstrate the high-fidelity, long-distance transmission of spatial patterns over unstabilized, 1 km multimode fibers using an adaptively trained neural network. Recent research on utilizing deep learning to reconstruct mixed spatial input using MMF has assumed either a stable transmission channel (e.g., very short MMF) or that a single network could generalize to all scenarios while sacrificing accuracy. This study presents a dynamic learning framework for tracking transmission model fluctuations, which led to extended MMFs with approximately 100% reconstruction accuracy. There are two features of the technique novelty: A multi-scale memory ensemble network that can compensate both slowly drifting system changes and abrupt distortions; and a self-supervised manner to fine tune the network during the continuous transmission process.

The writing in the manuscript is clear. Extensive demonstrations of the experimental results make them appear impressive and convincing. The authors also offer a fascinating use for compressively compressed video transmission, where errorless decoding is made possible thanks to the suggested approach's improved reconstruction performance, which significantly outperforms earlier research in data throughput. Since the descattering methodology is applicable to wide scattering circumstances and multimode fibers are promising platforms for light delivery in communication and imaging, there are probably many uses for the method presented here. Therefore, after taking into account the following criteria, I would that the article be published in Nature Communications.

Comments:

1. Could the authors explain why they design a multi-memory network with two branches representing interleaved short-term modeling and one branch for long-term modeling, instead of one short-term and one-long-term structure?

2. Could the author discuss more on the choice of self-supervised model updating interval? Is it relevant to the MMF characteristics only or also affected by other factors (e.g., imaging rate)?

3. In Fig. 2 and Fig. 4, the time axis of transmission accuracy trace starts from 500s. I understand that this is for comparison with the StaticNN baseline, but this could be a little bit misleading and incomplete. Could the authors reorganize their presentation (for example, add the first 500-second results as a Supplementary figure)? In addition, are the averaged spatial accuracy maps for MMDN calculated on 500s or 1000s duration?

4. What is the beam size of the incident light on the SLM and onto the entrance of the MMF? The authors have experimented on MMFs with different diameters. Is the beam size the same in all experiments and does the fill factor influence on the transmission performance?

5. I would suggest the authors to provide their code to facilitate the audience.

Minor comments:

the calibration of TM with huge number of -> the calibration of TM with a huge number of
after long time-> after a long time

with several orders throughput enhancement -> with several orders of throughput enhancement

For 100m transmission cases, MMDN even reduce the averaged error rate -> For 100m transmission cases, MMDN even reduces the average error rate

Reviewer #3 (Remarks to the Author):

Brief summary

Temporal dynamics of scattering media such as MMFs or biological tissues infamously hinders stable and high-quality image transmission and recovery without constant re-calibration. The authors propose an adaptive learning strategy which is capable of accounting for the different types of perturbations in real time. The article presents a self-supervised learning approach for binary image reconstruction through multimode fibers (MMFs) using a new model, the Multi-Scale Memory Dynamic-Learning Network (MMDN). The MMDN compensates for temporal dynamics of MMFs on different time scales, and has demonstrated consistent, long-term stability with over 99.9% image reconstruction accuracy for 1000 seconds.

The architecture of MMDN was explained and compared to StaticNN, its static counterpart. MMDN showed superior long-term stability, maintaining high accuracy over a longer period than StaticNN. A detailed analysis showed how MMDN utilized information from different timescales to maintain this stability. Subnetworks within MMDN made estimates based on varying temporal dynamics, and these estimates were combined according to their confidence scores to further improve accuracy.

Tests on various datasets demonstrated MMDN's broad applicability and potential for efficient adaptation via transfer learning. Finally, MMDN was used in a proof-of-concept video streaming

experiment through MMFs. A full-color video was encoded into a spatially multiplexed bit stream, and MMDN enabled a high-quality, high-throughput video stream, significantly outperforming StaticNN.

Areas of improvement

1) Would this work be of significance to the field? Currently shown evidence is not sufficiently convincing. A comparison with single-mode fiber (SMF), which is widely used in optical communication, could further underscore the significance of your findings. Evaluating the potential benefits of integrating the MMDN with MMFs in contrast to SMFs would be insightful. Specifically, if the SLM or DMD's speed is limited to certain kHz, how would the bandwidth of MMFs compare with that of SMFs?

2) Needs more convincing demos than binary input. While the video streaming experiment has been demonstrated to showcase its potentials to optical telecommunications, its impact on other applications such as remote sensing and endoscopy is unclear to me. It seems to me that MMDN is designed for decoding binary input patterns, while the images obtained in remote sensing and endoscopy are normally non-binary. Besides, MMDN requires real-time model correction via inspecting images at the distal end, which is arguably only possible with some types of "guidestars" as a feedback signal and thus substantially restricts its use scenario. On the other hand, data streaming via binary pattern decoding limits the operation of MMF at its maximum possible throughput, where a continuous level (or at least gray level) streaming is available. To validate its broad impact and applicability, it would be valuable to see some non-binary decoding results. Also, discussions on some real-world scenarios in which MMDN is viable would be helpful.

3) Needs more clarity on how it works. The primary merit of MMDN is its self-supervised adaptive learning capability. However, how the online learning works is neither elucidated in detail nor accompanied with proper reference. Without knowing the exact working principle and how the unseen unlabeled data are utilized to update the model, it is not clear whether MMDN serves as an efficient solution to perturbation-robust data transmission in MMFs.

4) Another concern is the amount of training data required for the initial model and each model update. First, 20,000 paired data for the initial model are not viable in most of the real applications in remote sensing and endoscopy even in a static system, which is probably even more challenging in a dynamic system as considered in the manuscript. In addition, it's not clearly stated in the manuscript how many patterns are needed for each update. Given the relatively small number of input modes and the development of efficient calibration methods (e.g., Li, S., Saunders, C., Lum, D.J. et al. Compressively sampling the optical transmission matrix of a multimode fibre. *Light Sci Appl* 10, 88 (2021).), if the number of patterns is comparable to the number of input modes, MMDN might not provide an edge on calibration efficiency, i.e., the number of data acquisitions required for each calibration. It would be useful to include more demonstration and discussion on the possibility of using a smaller training

dataset and its calibration efficiency compared to conventional methods such as transmission-matrix-based method.

5) Description of optical setup and quantification of MMF status.

a. The statement "We experimented on an amplitude-to-amplitude optical setup" requires clarification.

i. Is "amplitude-to-amplitude" synonymous with "intensity-to-intensity"?

ii. How does the SLM plane relate to the fiber input plane? Is the SLM on the Fourier plane of the objective lens, and are these two planes conjugate or Fourier related?

b. The characterization of the MMFs as "highly variable" in the experiment involving two different MMF channels needs to be better defined. As per Supplementary Figure 2, these MMF channels seem to fall under "medium instability".

Opinions

Overall, while the article develops a new network with interesting scattering results and analysis, there is not sufficient evidence that it provides a substantial breakthrough in the methods or significance in the application of optical communications.

Self-supervised dynamic learning for long-term high-fidelity image transmission through unstabilized diffusive media

Author response

We sincerely thank all reviewers for their helpful comments and suggestions, which have helped us improve the manuscript. We have made point-by-point responses below and have indicated where edits were made in the revised manuscript. We hope that the reviewers find our responses satisfactory and that the manuscript is now acceptable for publication in Nature Communications.

Reviewer 1

In this manuscript, Zhou et al. report a dynamic neural network for image transmission through diffusive MMFs. The merits of the reported method include the claimed 99.9% reconstruction accuracy, self-supervised learning with no need for labeled data, and the generalization to different image classes. However, the manuscript in its current form failed to explain the key points of their method, and the experimental setup might be too ideal to validate the performance of the reported network in real-world applications.

Response: We appreciate the reviewer's comments. Please find below our point-by-point responses.

Major points:

1. The most important part of the reported neural network, the self-supervised training, lacks essential details or explanations.

(a) It is not clear how the loss function is explicitly defined, and what regularization the authors used in network training.

(b) The convergence of the reported self-supervised training framework is questionable. The authors proposed to regard the previous network output as the label for current input data. But for the reported network architecture frequently updating by the input sequences, one can easily figure a trivial solution, in which the network produces constant for any input. The proof of correct convergence for the reported self-supervised training framework must be provided.

Response: We thank the reviewer for the helpful suggestion on explaining the proposed self-supervised learning.

(a) The loss function for each sub-network in the MMDN framework is the cross-entropy loss between the sub-network prediction and its 'ground-truth label' (that is, the digitized ensemble prediction using the previous models). The cross-entropy loss between predicted and ground-truth images is calculated pixel-by-pixel and

integrated. Assume the input speckle pattern to be S , the subnetwork prediction can be expressed as $f(S)$, $f = S1, S2, L3$, and the ‘ground-truth’ image is denoted as I . The loss function can be expressed as

$$CE(f, r) = - \sum_{i=0}^K I_r^i \log(f(S)_r^i),$$

$$CE(f) = \sum_{r=1}^N CE(f, r).$$

Here, $r = (x, y)$ is the spatial index of the image, K is the gray level for the intensity (e.g., $K=1$ for binary images), and i is the predicted value. A detailed schematic of how we calculate the loss and propagate the loss backward for network training is presented in Fig. R1. The deviations to be back-propagated come from two parts: 1) the adaptive fusion of three subnetworks gives an ensemble prediction that surpasses each sub-model; 2) the low-bit digitization acting like a nonlinear function helps to correct small errors.

Fig. R1 Schematic of the multi-timescale network training

To regularize the network training, we leverage the Dropout[1] to avoid over-fitting, and Batch Normalization[2] to scale the feature after each convolution layer. In our experiments, we set a dropout factor of 40%. We have also utilized early stopping to prevent overfitting in the update procedure. In addition, the low-bit

quantization also serves as a strong regularization. We have included the explanation of the loss function and regularization to the Neural network architecture paragraph in Methods and added Fig. R1 as Supplementary figure S4.

(b) We agree with the reviewer that a trivial solution in which the network produces constant for any input could fit our proposed dynamic framework. However, if the neural network is updated at a frequent pace, it will not easily fall into trivial solutions. Since strictly proving the convergence of a neural network in math is commonly infeasible, here we will offer some explanations from a conceptual standpoint.

- First, we would like to mention that the neural network is carefully designed, and is trained on a relatively large number of datasets during each update with the advanced optimizer and the Dropout, Batch Normalization, and early stop regularizations described in response (a). These operations are generally sufficient to ensure that the network converges.
- Second, since we properly set the time interval for network update during which period the MMF channels drift modestly, the network model optimally trained on previous dataset will exhibit only *minor* deviations from being 'optimal' for the current input data. Therefore, after initially training a generalized pre-train model, we *fine-tune* the network at the following update intervals rather than re-train the network from scratch. The previous model acting as the *pre-trained model can provide good initial parameter values to ensure the convergence of the network update*.
- In addition, we experimentally demonstrate the convergence of our MMDN by repeating the MMF spatial transmission experiments (fiber length: 1 km, core diameter: 200 μm) and analyzing the network convergence, as shown in Fig. R2. The first row shows the reconstruction accuracy at each time point. We randomly choose two time points and check the reconstructed images and training curves. We see that the MMDN generates desired reconstructed images for all three trials instead of a trivial constant solution after multiple rounds of updates. From the training curve, we further proved that the network training could reach convergence.
- Lastly, we try to discuss some cases where the network may encounter trivial solutions. One extreme case is when the input data remains unchanged for a considerably long period, leading the network to be trained on highly specialized datasets. Besides, reducing the amount of training data will increase the difficulty of network convergence.

We have added the discussion on network convergence to the Discussions in the revised manuscript.

Fig. R2 Examination of network training convergence in repeated experiments.

2. As a dynamic neural network, the update mechanism was hardly explained in the manuscript. How does the proposed CNN update its memory by its input sequence? What is the difference between S1, S2, and L3? What is the updating frequency for these three modules?

Response: We thank the reviewer for the helpful suggestion. We are sorry for not explaining the update mechanism clearly in our initial manuscript. We have revised Fig. 1b in the main text to give a detailed description of the dynamic learning pipeline, as also shown in Fig. R3 below. Overall, all three modules S1, S2, and L3 are updated at a short interval of 10s for all experiments via the proposed self-supervised scheme (see Fig. R3(c)). The update process is

essentially the fine-tuning of the network model using the new input data, while the pre-train model is trained on previous input data, and we refer to this process as ‘updating the network memory’. However, several critical issues of this naïve dynamic updating mechanism exist, including the vulnerability to abrupt channel deformations and that the network correction is expected to fail after long durations with the accumulation of previous errors. To tackle it, we introduce memory-wiping to modules S1 and S2 to let them refresh their memory (i.e., learn from scratch with the new input sequence) after several rounds of updates. Specifically, S1 and S2 modules are rebuilt alternatively at every five intervals (see Fig. R3(b)). This way, S1 and S2 only keep short-term memory while L3 keeps track of all the input sequences. The long-term module L3 has a more generalized performance, yet degrades inevitably after a long time (see Supplementary Fig. 4a for the performance of the L3-only model). The S1 and S2 draw more attention to the channel characteristics in the short term and can handle the sudden jitter effects that we observed in experiments.

Fig. R3 Pipeline of multi-timescale network update mechanism

We have added the presentation of network update mechanism and frequency in the Neural network architecture paragraph in Methods in the revised manuscript: “All three modules S1, S2, and L3 are frequently fine-tuned on the newly input sequence at a short interval (we chose 10s for all experiments) via the proposed self-supervised scheme. In addition, S1 and S2 modules are rebuilt alternatively at every five intervals to refresh their memory of the input data sequence and only keep short-term memory.”

3. The experimental setup seems too ideal for the validation of real-world applications. The inputs and the targets of the network were both binary images, which greatly reduces the difficulty of this reconstruction task and hinders

the potent of the reported method for practical applications. The authors should at least show results on non-binary images, e.g., 2-bit, 4-bit images.

Response: We thank the reviewer for the helpful suggestion. We agree with the reviewer that the binary image transmission reduces the difficulty of reconstruction. To validate the potential of our proposed method in real-world applications, we extend MMDN to transmit non-binary images. Specifically, we tested on 2-bit and 4-bit encoded random images. Experimenting on 1km length step-index MMF (core dia. 200 μm), we demonstrated long-term transmission of 2-bit images with 24 \times 24-pixel (see Fig. R4(a)) and 4-bit images with 16 \times 16-pixel (see Fig. R4(b)).

Fig. R4 Performance of non-binary encoded image transmission over long time in unstabilized MMF.

The mean absolute errors (MAEs) of recovered non-binary patterns are measured in temporal and spatial domains. The 1-MAE metric is above 0.994 for 2-bit reconstruction and 0.992 for 4-bit reconstruction during the 500-second period, and the average accuracy is 0.998 and 0.999, respectively. As a comparison, the static neural network trained on 20,000 datasets will intensely degrade in reconstruction accuracy. The experiments proved the potential of the proposed MMDN approach for practical applications involving gray-level image delivery. **We have added the non-binary encoding results to the Long-term spatial transmission through unstabilized MMFs subsection in Results in the revised manuscript.**

4. In addition to the static neural network baseline, I also wonder how conventional algorithms would perform against the reported method. Benchmarking would be good to include.

Response: We thank the reviewer for the helpful suggestion. Transmission matrix (TM) based algorithms have been widely investigated in previous works to solve the de-scattering problem in MMF. Here, we added a comparison of our proposed MMDN with TM-based reconstruction. We adopted a phase retrieval-based TM calibration approach using binary DMD for amplitude modulation[3] to measure the complex TM of the scattering MMF, showcased in Fig. R5(a). The image reconstruction is achieved by solving the phase retrieval problem with the Gerchberg-Saxton (GS) algorithm given the calibrated TM and captured intensity distribution.

We reproduced the experiments of long-term transmission in Fig.2. A set of $4N$ input-speckle data pairs (N is the pixel number of input patterns) captured right before the testing data is used for TM calibration. The reconstruction accuracy for various MMF systems using MMDN and TM-based approaches is compared in Fig. R5. We see that the accuracy of using TM-based reconstruction is mostly below 99%, lower than the neural network-based methods, and will gradually reduce due to the system drifts. **We have added the TM-based benchmark algorithm for comparison in Supplementary Fig. 6 in the revised manuscript.**

Fig. R5 Comparison in reconstruction accuracy between MMDN and TM-based algorithms. (a) Calibrated TM for 100m and 1km MMFs, measured with 1024 pairs of 16x16-pixel input images and 100x100-pixel output speckles. (b-e) Transmission accuracy for 500s duration experimented on various MMFs.

Minor points:

1. The specs of the MMF used in the experiment are not listed.

Response: We thank the reviewer for the comment, we have added a supplementary table to summarize the specifications of the MMs used in all experiments.

Specifications	Fig. 2a	Fig. 2b	Fig. 2c	Fig. 2d	Fig. 3a	Fig. 3b	Fig. 4 a,b	Fig. 4 c,d	Fig. 5-6
Type	Step-Index				Step-Index		Gradient-Index	Step-Index	Step-Index
Core diameter	200 μm				200 μm		200 μm	200 μm	200 μm
Length	1 km		100 m		1 km		100 m	1 km	1 km
Image res	32 \times 32	16 \times 16	32 \times 32	16 \times 16	24 \times 24	16 \times 16	16 \times 16	32 \times 32	32 \times 32
Grayscale	1-bit				2-bit	4-bit	1-bit		1-bit

Specifications	Fig. S2							
Type	Step-Index	Step-Index	Gradient-Index	Step-Index	Step-Index	Step-Index	Step-Index	Step-Index
Core diameter	200 μm	200 μm	200 μm	50 μm	200 μm	50 μm	200 μm	200 μm
Length	100 m	100 m	100 m	100 m	1 km	1 km	1 km	1 km
Image res	16 \times 16	32 \times 32	16 \times 16	16 \times 16	16 \times 16	16 \times 16	16 \times 16	32 \times 32
Grayscale	1-bit							

Specifications	Fig. S3	Fig. S5 a,b	Fig. S5 c,d	Fig. S6b	Fig. S6c	Fig. S6d	Fig. S6e	Fig. S8
Type	Step-Index							
Core diameter	200 μm	100 μm	200 μm	200 μm				200 μm
Length	1 km	1 km		1 km	100 m	1 km	100 m	1 km
Image res	32 \times 32	16 \times 16	32 \times 32	32 \times 32		16 \times 16		32 \times 32
Grayscale	1-bit							

2. In line 219, please give a reference to AdaDelta paper.

Response: We have added a reference to the AdaDelta paper in our revised manuscript: Matthew D. Zeiler. "Adadelta: an adaptive learning rate method." arXiv preprint arXiv:1212.5701 (2012).

3. The number of trainable parameters and the training time on the given machine are not mentioned in the Methods section.

Response: We thank the reviewer for the comment. The neural network we developed consists of two convolutional layers both with 8 convolution kernels of 3*3 kernel size and stride 2, and one fully-connected layer to map the feature map to the dimension of the target pattern. The trainable parameter numbers are 72 and 576 for the two convolutional layers. For $N \times N$ -pixel image transmission with input speckle dimension of $M \times M$ -pixel, the number of parameters for the fully connected layer is $8 * (M/2)^2 * N^2$. The training time for each round of network update processed on the workstation equipped with NVidia RTX3090 GPU is around 0.11 seconds per epoch for the 16×16 -pixel images and 0.98 seconds for 32×32 -pixel images. We adopt the early stopping to automatically stop the training, and the network fine-tuning typically can be finished within 20 epochs. **We have provided a detailed description of the trainable parameters number and the training time in the Methods section.**

4. For Fig. 1b, the pink texts on the left, what is “self-s update”?

Response: We are sorry for the misleading spelling. The text “self-s update” is the abbreviation for “self-supervised update”. **To make it clearer, we have changed the texts to “self-supervised update” in Fig. 1b.**

Reviewer 2

The authors demonstrate the high-fidelity, long-distance transmission of spatial patterns over unstabilized, 1 km multimode fibers using an adaptively trained neural network. Recent research on utilizing deep learning to reconstruct mixed spatial input using MMF has assumed either a stable transmission channel (e.g., very short MMF) or that a single network could generalize to all scenarios while sacrificing accuracy. This study presents a dynamic learning framework for tracking transmission model fluctuations, which led to extended MMFs with approximately 100% reconstruction accuracy. There are two features of the technique novelty: A multi-scale memory ensemble network that can compensate both slowly drifting system changes and abrupt distortions; and a self-supervised manner to fine tune the network during the continuous transmission process.

The writing in the manuscript is clear. Extensive demonstrations of the experimental results make them appear impressive and convincing. The authors also offer a fascinating use for compressively compressed video transmission, where errorless decoding is made possible thanks to the suggested approach's improved reconstruction performance, which significantly outperforms earlier research in data throughput. Since the descattering methodology is applicable to wide scattering circumstances and multimode fibers are promising platforms for light delivery in communication and imaging, there are probably many uses for the method presented here. Therefore, after taking into account the following criteria, I would that the article be published in Nature Communications.

Response: We appreciate the reviewer's comments. Please find below our point-by-point responses.

Comments:

1. Could the authors explain why they design a multi-memory network with two branches representing interleaved short-term modeling and one branch for long-term modeling, instead of one short-term and one-long-term structure?

Response: We thank the reviewer for the comment. In general, the ensemble of multiple submodules gives superior performance over the single-module network, and the improvement will increase with the number of submodules. Therefore, building two short-term memory models is promising to outperform the one short-term design. A more critical reason that we design the multi-memory network with two branches representing interleaved short-term modeling and one branch for long-term modeling is that the short-term model is less stable than the long-term model. By introducing the frequent rebuild to the short-term model, we wipe its memory and let it emphasize adjacent data. However, since the model is rebuilt on a relatively smaller volume of dataset compared to the long-term model, it is prone to have low generalization. To alleviate this problem, we propose to construct two short-term models that rebuild alternatively. This way, when one short-term model is rebuilt, the other short-term model has been updated for several rounds and owns high generalization. It should be noted that further increasing the number of submodules rises the computational budget, hence we stay with two short-term memory models to balance the computation accuracy and efficiency.

To experimentally demonstrate the superiority of the three-branch architecture over the two-branch one, we conduct three implementations of MMDN: (i) ensemble learning of all three submodules, noted as S1+S2+L3, (ii) ensemble learning of S1 and L3, noted as S1+L3 and (iii) ensemble learning of S2 and L3, noted as S2+L3. Comparing the reconstruction accuracy in both spatial and temporal domains, as shown in Fig. R6, the S1+S2+L3 model outperforms the S1+L3 and S3+L3 models.

Fig. R6 Comparison in reconstruction accuracy of different MMDN implementations.

2. Could the author discuss more on the choice of self-supervised model updating interval? Is it relevant to the MMF characteristics only or also affected by other factors (e.g., imaging rate)?

Response: We thank the reviewer for the comments. In our experiments, an updating interval of 10s is chosen for each sub-networks, and the two short-term memory models will be alternatively rebuilt after 100s. The choice of update interval mainly considers two factors: the MMF characteristics change speed and the computational cost for model re-training. Referring to the stability calibration in Supplementary Fig. 2, we see that a static model depicting the MMF channels would quickly degrade after 10~20s. Therefore, we set the model update interval to be 10s during which the neural network could produce a reliable prediction. Setting a smaller interval can work, yet will increase the computation cost through frequent training. The imaging rate does not affect the choice of updating interval in general. Whereas, if the imaging rate is too low, there might not be enough amount of data for the network update.

In this case, using longer updating intervals might help. We have included the updating interval to the main text in the Multi-scale memory dynamic-learning network subsection.

3. In Fig. 2 and Fig. 4, the time axis of transmission accuracy trace starts from 500s. I understand that this is for comparison with the StaticNN baseline, but this could be a little bit misleading and incomplete. Could the authors reorganize their presentation (for example, add the first 500-second results as a Supplementary figure)? In addition, are the averaged spatial accuracy maps for MMDN calculated on 500s or 1000s duration?

Response: We thank the reviewer for the comment. We agree with the reviewer that the transmission accuracy trace does not cover the complete 1000s duration in Fig. 2 and Fig. 4. Actually, a complete trace is provided in Supplementary Fig. 4 for comparison of single-scale and multi-scale memory dynamic learning approaches. From the two example MMF channels covering both high and medium instability scenarios, we can see that the accuracy in the previous 500s is preserved at a high level. On the other hand, if the reconstruction accuracy is quite low in the previous 500s duration, the error will accumulate during the dynamic update procedure. Therefore, presenting the results from the latter 500s period would be reasonable in the authors' opinion.

The averaged spatial accuracy maps for MMDN in Fig. 2 and 4 are both calculated on the latter 500s duration for a fair comparison with the StaticNN baseline. We have added the information in the main text in the revised manuscript.

4. What is the beam size of the incident light on the SLM and onto the entrance of the MMF? The authors have experimented on MMFs with different diameters. Is the beam size the same in all experiments and does the fill factor influence on the transmission performance?

Response: We thank the reviewer for the comment. The beam size of the incident light on the SLM has a $1/e^2$ diameter of 7 mm, and an effective area of 384×384 -segment on the SLM is used to modulate the input light field. The reflected light beam has a diameter of ~ 5 mm and is focused by the objective to couple into the fiber. Different from the designs that use a 4f-system to conjugate the spatial profile on the SLM plane to the fiber input plane, in our setup, the input beam is tightly focused at the fiber input plane, regardless of the MMF core diameters. As the experiments demonstrated, our proposed algorithm shows good transmission performance in different MMF settings. We have revised the description for optical setup in the Experimental set-up paragraph in the Method section.

5. I would suggest the authors to provide their code to facilitate the audience.

Response: We appreciate the reviewer's comment. The code for MMDN has been provided in our revised submission with a GitHub link for review and to the audience.

Minor comments:

the calibration of TM with huge number of -> the calibration of TM with a huge number of

after long time-> after a long time

with several orders throughput enhancement -> with several orders of throughput enhancement

For 100m transmission cases, MMDN even reduce the averaged error rate -> For 100m transmission cases, MMDN even reduces the average error rate

Response: We thank the reviewer for pointing these typos out. We have changed all the texts to the correct formats.

- the calibration of TM with huge number of -> the calibration of TM with *a* huge number of
- after long time-> after *a* long time
- with several orders throughput enhancement -> with several orders *of* throughput enhancement
- For 100m transmission cases, MMDN even reduce the averaged error rate -> For 100m transmission cases, MMDN even *reduces* the *average* error rate

Reviewer 3

Brief summary

Temporal dynamics of scattering media such as MMFs or biological tissues infamously hinders stable and high-quality image transmission and recovery without constant re-calibration. The authors propose an adaptive learning strategy which is capable of accounting for the different types of perturbations in real time. The article presents a self-supervised learning approach for binary image reconstruction through multimode fibers (MMFs) using a new model, the Multi-Scale Memory Dynamic-Learning Network (MMDN). The MMDN compensates for temporal dynamics of MMFs on different time scales, and has demonstrated consistent, long-term stability with over 99.9% image reconstruction accuracy for 1000 seconds.

The architecture of MMDN was explained and compared to StaticNN, its static counterpart. MMDN showed superior long-term stability, maintaining high accuracy over a longer period than StaticNN. A detailed analysis showed how MMDN utilized information from different timescales to maintain this stability. Subnetworks within MMDN made estimates based on varying temporal dynamics, and these estimates were combined according to their confidence scores to further improve accuracy.

Tests on various datasets demonstrated MMDN's broad applicability and potential for efficient adaptation via transfer learning. Finally, MMDN was used in a proof-of-concept video streaming experiment through MMFs. A full-color video was encoded into a spatially multiplexed bit stream, and MMDN enabled a high-quality, high-throughput video stream, significantly outperforming StaticNN.

Response: We appreciate the reviewer's comments. Please find below our point-by-point responses.

Areas of improvement

1) Would this work be of significance to the field? Currently shown evidence is not sufficiently convincing. A comparison with single-mode fiber (SMF), which is widely used in optical communication, could further underscore the significance of your findings. Evaluating the potential benefits of integrating the MMDN with MMFs in contrast to SMFs would be insightful. Specifically, if the SLM or DMD's speed is limited to certain kHz, how would the bandwidth of MMFs compare with that of SMFs?

Response: We thank the reviewer for the valuable comments. The main contribution of this work is that we develop an artificial intelligence aided image reconstruction approach to empower accurate spatial-multiplexed information transmission through MMF over long distance and long time. Although SMF has been widely used in optical communication, the investigation on MMF has drawn increasing interest for its advantage in preserving spatial information and shows potential in applications such as:

1. High-capacity transmission: the MMF allows spatial-division multiplexing to promote transmission capacity. Regarding the reviewer's particular concern about the bandwidth in our proof-of-concept system, we agree with the reviewer that the current setup using DMD with 22 kHz modulation speed will limit the bandwidth of MMF transmission to < tens of MHz. One promising and practical solution to break this limit is to use modulated lighting

arrays as the spatial information encoder. Independently addressable LED arrays of 10×10 pixels with 100 MHz switching speed[5] and 64×64 pixels with 2.5 MHz[6] have been reported to generate fast-switching structured illumination patterns at low cost. Using a laser diode array, for example, the VCSEL array, which is used in LiDAR applications and data centers, spatial modulation with up to Gigahertz bandwidth can be achieved. Equipping with those high-speed spatial modulation modules, the bandwidth of MMFs could be elevated to hundreds of Gigahertz. With the combination of multi-level modulation and multi-wavelength multiplexing, the transmission capacity could be further expanded. Therefore, we believe that integrating the MMDN with MMFs to enable spatial-multiplexed multi-input multi-output (MIMO) transmission in a compact thin channel has potential benefits compared to SMFs, as the single-input transmission mechanism reaches its bandwidth limit. We will develop high capacity MMF transmission using high bandwidth laser transmitter array for future work.

2. Direct image transmission: the large core of MMF enables the propagation of over thousands of spatial modes, thus can encode image information. This allows the delivery of high degree-of-freedom light field information through the MMF channel[4][14]. With the proposed MMDN technique, one could unscramble the speckle output caused by spatial mode coupling and retrieve high-resolution image information or even semantic information without recovering the image. On the other hand, the potential applications of our proposed work are not limited to optical communication, but can be used for remote imaging and endoscopic imaging. Compared with previous methods that commonly use wavefront shaping to focus the output field into a raster scanning spot, our MMDN approach shows the benefit of direct image formation *from single capture* that can rapidly operate at kilohertz framerates. Using a fast camera could further promote the imaging speed by several orders. We explored the feasibility of MMDN for both binary images and non-binary encoded pattern recovery (see the response for comment 2)), revealing its potential for remote imaging.
3. Secure transmission: the inter-mode coupling nature of the MMF makes it a nonlinear scattering channel that intrinsically encode the input information. Hence, MMF holds a promise for optical secure transmission and quantum cryptography. Previous literatures have reported on MMF based quantum information transmission [13][14]. Applying our proposed MMDN to spatial-division optical encryption, we could further enhance the spatial dimensionality to over a thousand and maintain a strong robustness to system instability.

To conclude, as a proof-of-concept demonstration of our image reconstruction approach, we validated the concept of MMDN by encoded video transmission. Nevertheless, given the potential applications of the MMF based image transmission system for the next generation high-capacity optical communication, direct remote imaging and information sensing, we believe this work would be of significance to the field. **We have discussed on the potential benefits and significance of MMF with MMDN in the Discussions section in the revised manuscript.**

2) Needs more convincing demos than binary input. While the video streaming experiment has been demonstrated to showcase its potentials to optical telecommunications, its impact on other applications such as remote sensing and endoscopy is unclear to me. It seems to me that MMDN is designed for decoding binary input patterns, while the images obtained in remote sensing and endoscopy are normally non-binary. Besides, MMDN requires real-time model correction via inspecting images at the distal end, which is arguably only possible with some types of

“guidestars” as a feedback signal and thus substantially restricts its use scenario. On the other hand, data streaming via binary pattern decoding limits the operation of MMF at its maximum possible throughput, where a continuous level (or at least gray level) streaming is available. To validate its broad impact and applicability, it would be valuable to see some non-binary decoding results. Also, discussions on some real-world scenarios in which MMDN is viable would be helpful.

Response: We thank the reviewer for the valuable comments.

- Non-binary decoding: We conducted experiments on non-binary input spatial transmission. Specifically, we tested on 2-bit and 4-bit encoded random images. Experimenting on 1km length step-index MMF (core dia. 200 μm), we demonstrated long-term transmission of 2-bit images with 24 \times 24-pixel and 4-bit images with 16 \times 16-pixel, as shown below. The mean absolute errors (MAEs) of recovered non-binary patterns are measured in temporal and spatial domains. The 1-MAE metric is above 0.994 for 2-bit reconstruction and 0.992 for 4-bit reconstruction during the 500-second period, and the average accuracy is 0.998 and 0.999, respectively. As a comparison, the static neural network trained on 20,000 datasets will intensely degrade in reconstruction accuracy. The implemented 2- and 4-bit gray-level encoding experiments improved the data throughput by 4- and 16-fold through the MMF channels, and proved the potential for practical applications involving gray-level image delivery and high-level modulated optical communication. We have added the non-binary encoding results to the Long-term spatial transmission through unstabilized MMFs subsection in Results in the revised manuscript. Even so, we should admit that the MMDN is not suitable for high-level encoded images because the regularization that constrains the recovered images to be binary or low-level grayscale becomes invalid. To address this issue, we need to design new regularizations for gray-level nature images, such as the minimization of spatial gradients[9] and high-level perception loss[10]. We will explore this for future work.

Fig. R7 Performance of non-binary encoded image transmission over long time in unstabilized MMFs.

- Model correction feasibility: First, we would like to point out that here MMDN only require the access to the distal end of the fiber in the pre-training stage. While for the dynamic network updating stage, the MMDN used the network predicted input patterns rather than the actual inspected images at the distal end as the ground-truth label. This means that the network prediction itself acts as a feedback signal and no extra “guide-stars” or feedback signal is required in the updating stage for model correction. Regarding to the practical application in endoscopy imaging scenarios, where the distal end is placed near the surface of or even buried in the biological tissue, one could first use a programmable spatial modulation device placed at the sample plane for system

calibration, then change back to the actual test sample. This pre-calibration procedure is a common operation reported in previous implementations of MMF-based imaging (e.g., Vasquez-Lopez et al. *Light: Science & Applications* (2018) 7:110 and Turtaev Sergey et al. *Light: Science & Applications* 7.1 (2018): 92). If the sample switching takes less than several seconds and the fiber is held mostly static, the pre-trained model would still work with slight performance deviation. In addition, the deviation can be compensated to some extent as the network dynamically updates.

3) Needs more clarity on how it works. The primary merit of MMDN is its self-supervised adaptive learning capability. However, how the online learning works is neither elucidated in detail nor accompanied with proper reference. Without knowing the exact working principle and how the unseen unlabeled data are utilized to update the model, it is not clear whether MMDN serves as an efficient solution to perturbation-robust data transmission in MMFs.

Response: We thank the reviewer for the comments. We are sorry for not explaining the online learning mechanism clearly in our initial manuscript. **We have revised Fig. 1b in the main text to give a detailed description of the dynamic learning pipeline**, as also shown in Fig. R8 below. Overall, all three modules S1, S2, and L3 are updated at a short interval (we chose 10s for all experiments) via the proposed self-supervised scheme (see Fig. R8(c)). The update process is essentially the fine-tuning of the network model which is trained on previous data with the new input data. However, several critical issues of this naïve dynamic updating mechanism exist, including the vulnerability to abrupt channel deformations and that the network correction is expected to fail after long durations with the accumulation of previous errors. To tackle it, we introduce memory-wiping to modules S1 and S2 to let them forget the previous model and learn from scratch with the new input sequence after several rounds of updates. Specifically, S1 and S2 modules are rebuilt alternatively at every five intervals (see Fig. R8(b)). This way, S1 and S2 only keep short-term memory while L3 keeps track of all the input sequences. **We have added the presentation of network update mechanism and frequency in the Neural network architecture paragraph in Methods in the revised manuscript**: “All three modules S1, S2, and L3 are frequently fine-tuned on the newly input sequence at a short interval (we chose 10s for all experiments) via the proposed self-supervised scheme. In addition, S1 and S2 modules are rebuilt alternatively at every five intervals to refresh their memory of the input data sequence and only keep short-term memory.”

Regarding the training procedure for one round of network update, the cross-entropy loss between the predicted image and the ‘ground-truth’ is calculated for each sub-network to let them adapt to the gradually drifting channel characteristics. To regularize the network training, we leverage the Dropout and early stopping to avoid over-fitting and Batch Normalization to scale the feature after each convolution layer. **We provided a Supplementary figure to illustrate the detailed network training for one interval**, as also shown below in Fig. R9.

Fig. R8 Pipeline of multi-timescale network update mechanism

Fig. R9 Schematic of the multi-timescale network training

4) Another concern is the amount of training data required for the initial model and each model update. First, 20,000 paired data for the initial model are not viable in most of the real applications in remote sensing and endoscopy even in a static system, which is probably even more challenging in a dynamic system as considered in the manuscript. In addition, it's not clearly stated in the manuscript how many patterns are needed for each update. Given the relatively small number of input modes and the development of efficient calibration methods (e.g., Li, S., Saunders, C., Lum, D.J. et al. Compressively sampling the optical transmission matrix of a multimode fibre. *Light Sci Appl* 10, 88 (2021).), if the number of patterns is comparable to the number of input modes, MMDN might not provide an edge on calibration efficiency, i.e., the number of data acquisitions required for each calibration. It would be useful to include more demonstration and discussion on the possibility of using a smaller training dataset and its calibration efficiency compared to conventional methods such as transmission-matrix-based method.

Response: We thank the reviewer for the comments.

- First, we would like to clarify that only in the pre-training stage do we need a large quantity of paired data, and the MMF is not required to maintain static during this procedure. This is because we use a neural network to model the transmission system, which offers higher robustness and generalization than transmission-matrix-based approaches on environmental instabilities[11]. Second, as explained in Response 2), both facets of the MMF are accessible in the pre-training stage, and if we replace the real sample with a programmable spatial light modulator at the distal end, acquiring a large set of paired data will be feasible. Actually, previous works using neural networks for image reconstruction followed a similar procedure to collect over several tens of thousands pairs of training set for accurate training (e.g., Piergiorgio Caramazza et. al. in *Nature Communications* 2019[4], and Rahmani et al. in *Light: Science & Applications* 2018[12]).
- For each update, we use P=500 paired data for N=16×16-pixel image reconstruction and P=1000 paired data for N=32×32-pixel image reconstruction. The required number of patterns is comparable to the number of input modes. **We have added the quantity of training dataset in the Neural Network paragraph in the Method section.** Considering recently reported efficient calibration methods such as *Light Sci. Appl.* 10, 88 (2021), one intuitive limitation of this approach is that it poses a strong assumption of the sparsity of TM in certain basis (i.e., weak coupling between the basis), which is only applicable for short fibers. For transmission over 100m length MMFs, the level of modal coupling will dramatically increase and the sparsity prior cannot be satisfied. As a comparison, our proposed MMDN is a general method that can work in both weak and strong coupling regimes. More importantly, even though the compressive methods allow efficient TM calibration, they still require extra measurements with known labels as the reference; while our MMDN dynamically corrects the system characteristics with the predicted testing data. That is, the input data sequence is mutually used for image reconstruction and recalibration, and the transmission efficiency is not affected. From this perspective, the calibration sampling rate for MMDN is 0% in terms of extra calibration pattern number.
- We further provide experiments on reconstruction performance using smaller training datasets. For the pre-

training stage, we acquired 200-second 32×32 -pixel data and randomly chose 20,000, 10,000, and 5,000 samples for training. We compute the accuracy on a validation set containing other 1000 samples, as shown in Fig. R10(a). From the results, we see that the accuracy of the pre-trained model decreases with a reduced training set. Similar experiments are conducted for 16×16 -pixel imaging, and the training set contains 5,000, 2,500, and 1,250 samples, respectively, see Fig. R10(b). To obtain a generalized and accurate initial model, we still fixed with 20000 paired data for 32×32 -pixel imaging and 5000 paired data for 16×16 -pixel imaging.

Fig. R10 Pre-train model accuracy with different sizes of training set (a) 32×32 -pixel (b) 16×16 -pixel. Fiber: SI, 1 km, core diameter $200 \mu\text{m}$, NA 0.22.

We also tested using a smaller dataset for each update. For 32×32 -pixel imaging, network update on a training set of 1000 and 500 paired data are tested, and the reconstruction accuracy in a 200s duration is shown in Fig. R11(a). For 16×16 -pixel imaging, 500 and 250 paired data are tested for each update, as shown in Fig. R11(a). Similarly, the reduction of data numbers will slightly reduce the accuracy at each update.

Fig. R11 Reconstruction accuracy at each update with different sizes of training set (a) 32×32 -pixel (b) 16×16 -pixel. Fiber: SI, 1 km, core diameter $200 \mu\text{m}$, NA 0.22.

5) Description of optical setup and quantification of MMF status.

- a. The statement "We experimented on an amplitude-to-amplitude optical setup" requires clarification.
- Is "amplitude-to-amplitude" synonymous with "intensity-to-intensity"?
 - How does the SLM plane relate to the fiber input plane? Is the SLM on the Fourier plane of the objective lens, and are these two planes conjugate or Fourier related?
- b. The characterization of the MMFs as "highly variable" in the experiment involving two different MMF channels needs to be better defined. As per Supplementary Figure 2, these MMF channels seem to fall under "medium instability".

Response: We thank the reviewer for the comments.

- a- i: The statement "amplitude-to-amplitude" is synonymous with "intensity-to-intensity", meaning that intensity-modulated images are coupled into the fiber and their intensity distributions at the distal end are measured. To make the concept clearer, **we have modified the statement in the main text in the Result Section**: We experimented on an intensity modulated optical setup.
- a- ii: Most reported MMF transmission systems placed the SLM at the conjugate plane of the fiber input plane with a 4f system consisting of a tube lens and an objective. However, this setup often requires a high magnification objective to achieve large magnification. In this work, we placed the SLM conjugated to the back-pupil plane of the objective by a pair of relay lenses of 1:1 magnification factor, and the fiber input plane is placed at the imaging plane of the objective. **We have included the description of the optical setup in the Experimental set-up paragraph in the Method section**.
- b- We would like to thank the reviewer for pointing this out. The two MMF channels in Fig. 3 should be categorized as "medium instability". **We have modified the texts in the main text in the Result section**: "We chose two highly variable MMF channels with medium instability including a 100m-length gradient-indexed MMF of 200 μm diameter (GI-200) and a 1km-length step-indexed MMF of 200 μm diameter (SI-200)".

Opinions

Overall, while the article develops a new network with interesting descattering results and analysis, there is not sufficient evidence that it provides a substantial breakthrough in the methods or significance in the application of optical communications.

Response: We sincerely thank the reviewer again for the valuable suggestions. We hope the revised manuscript is acceptable to the reviewer.

References

- [1] Hinton, Geoffrey E., et al. Improving neural networks by preventing co-adaptation of feature detectors. arXiv preprint arXiv:1207.0580, 2012.

- [2] Ioffe, Sergey, and Christian Szegedy. Batch normalization: Accelerating deep network training by reducing internal covariate shift. International conference on machine learning. pmlr, 2015.
- [3] Drémeau, Angélique, et al. "Reference-less measurement of the transmission matrix of a highly scattering material using a DMD and phase retrieval techniques." *Optics express* 23(9): 11898-11911, 2015.
- [4] Caramazza P, Moran O, Murray-Smith R, et al. Transmission of natural scene images through a multimode fibre. *Nature Communications*, 10(1): 2029, 2019.
- [5] Wang M, Sun M, Huang C. Single-pixel 3D reconstruction via a high-speed LED array. *Journal of Physics: Photonics*, 2(2): 025006, 2020.
- [6] Zhao W, Chen H, Yuan Y, et al. Ultrahigh-speed color imaging with single-pixel detectors at low light level. *Physical Review Applied*, 12(3): 034049, 2019.
- [7] Vasquez-Lopez et al. High-fidelity multimode fibre-based endoscopy for deep brain in vivo imaging. *Light: Science & Applications*, 7:110, 2018.
- [8] Turtaev, Sergey, et al. Subcellular spatial resolution achieved for deep-brain imaging in vivo using a minimally invasive multimode fiber. *Light: Science & Applications* 7.1: 92, 2018.
- [9] Rudin, Leonid I., Stanley Osher, and Emad Fatemi. Nonlinear total variation based noise removal algorithms. *Physica D: nonlinear phenomena* 60.1-4: 259-268, 1992.
- [10] Johnson, Justin, Alexandre Alahi, and Li Fei-Fei. Perceptual losses for real-time style transfer and super-resolution. *European Conference in Computer Vision*, October 11-14, 2016.
- [11] Li Y, Xue Y, Tian L. Deep speckle correlation: a deep learning approach towards scalable imaging through scattering media. *Optica*, 5: 1181-1190, 2018.
- [12] Rahmani B, Loterie D, Konstantinou G et al. Multimode optical fiber transmission with a deep learning network. *Light: Science & Applications*, 7: 69, 2018.
- [13] Defienne, H., Barbieri, M., Walmsley, I. A., Smith, B. J. & Gigan, S. wo-photon quantum walk in a multimode fiber. *Science Advances*, 2: 1e1501054, 2016.
- [14] Zhou, Y., Braverman, B., Fyffe, A. et al. High-fidelity spatial mode transmission through a 1-km-long multimode fiber via vectorial time reversal, *Nature Communications*, 12: 1866, 2021.

REVIEWER COMMENTS

Reviewer #1 (Remarks to the Author):

I think the manuscript has been improved during the revision process and most of the referee comments have been well addressed.

Reviewer #2 (Remarks to the Author):

I am happy with the revision of the manuscript and recommend acceptance as it is.

Reviewer #3 (Remarks to the Author):

The authors have effectively addressed some of the concerns. The supplementary videos are impressive. But I hope the authors can make the paper a bit more rigorous. There are a few remaining concerns.

1. In the first round of revision, we felt it is important to do a direct comparison with single-mode fibers to demonstrate significance and thus requested the experiments. But the authors provided only the discussion on the advantages of the MMFs without supporting data.

2. From response: "First, we would like to clarify that only in the pre-training stage do we need a large quantity of paired data, and the MMF is not required to maintain static during this procedure. This is because we use a neural network to model the transmission system, which offers higher robustness and generalization than transmission-matrix-based approaches on environmental instabilities[11]."

a. This is a good reference to compare. But the description is incorrect. On the contrary, reference 11 in response does not use transmission-matrix-based approaches. It actually uses very similar approaches to the presented work except it's for dynamic diffuser rather than dynamic MMF.

b. How is this work compared to reference 11 in response and reference 18 in paper, in terms of innovation and performance?

3. Rigor of paper writing, I didn't do a thorough check, but noticed that a few obvious citation errors. Please correct throughout. Here are the two instances that I noticed in main text:

- a. Reference 7 and 10 are the same one
- b. References 18 and 25 are the same one

4. As a minor point, I would suggest including Fig. R10 and R11 in the supplementary (surrounded by a proper discussion), which serve as a valuable reference for the audience who are interested in deploying the proposed method while the scarcity of training data is of concern.

Self-supervised dynamic learning for long-term high-fidelity image transmission through unstabilized diffusive media

Author response

We sincerely thank all reviewers for their helpful comments and suggestions, which have helped us improve the manuscript. Regarding the concerns from Reviewer 3, we have made point-by-point responses below and indicated where edits were made in the revised manuscript. We hope that the reviewers find our responses satisfactory and that the manuscript is now acceptable for publication in *Nature Communications*.

Reviewer 1

I think the manuscript has been improved during the revision process and most of the referee comments have been well addressed.

Response: We appreciate the reviewer's valuable help for improving our manuscript in the first round of revision.

Reviewer 2

I am happy with the revision of the manuscript and recommend acceptance as it is.

Response: We appreciate the reviewer's valuable help for improving our manuscript in the first round of revision.

Reviewer 3

The authors have effectively addressed some of the concerns. The supplementary videos are impressive. But I hope the authors can make the paper a bit more rigorous. There are a few remaining concerns.

Response: We appreciate the reviewer's comments. Please find below our point-by-point responses.

1. In the first round of revision, we felt it is important to do a direct comparison with single-mode fibers to demonstrate significance and thus requested the experiments. But the authors provided only the discussion on the advantages of the MMFs without supporting data.

Response: In this round of revision, we have followed the reviewer's suggestion and experimentally conveyed some direct comparison of our proposed method with SMF-based data transmission.

- First of all, we would like to emphasize again that the essential advantage of our MMF imaging system with MMDN is that it can achieve direct imaging and transmission of a 2D optical field by one-shot. On the contrary, if we use a single-mode fiber for imaging, a parallel-to-serial conversion procedure is required. Specifically, one representative pipeline follows a raster-scanning mechanism, i.e., it adopts a two-dimensional scanning module and transmits the optical field pixel-by-pixel; the other pipeline first records the spatial information via a camera and transmits the serialized signals through the SMF. As a result, the SMF-based imaging systems are limited in speed by either the slow mechanical scanning or the opto-electro-optical conversion and signal processing.

Figure R1. Illustration of MMF and SMF-based imaging techniques

To experimentally compare the rapid imaging performance using MMF and SMF, we build the imaging systems for cases i) and ii) and adopted the DMD switching at up to 20kHz to emulate the dynamic scene at the distal end, as shown in Figure R2. By using an ultrafast camera (Photron FASTCAM SA-Z, >100k fps @ 256×256-resolution) for speckle image capturing at the MMF output, we achieved 20k fps spatial imaging. For SMF-based imaging, we built a fast 2D scanning setup by using a pair of galvo-scanners (Thorlabs, <1000 Hz scanning speed) and the scanning speed is ~100Hz per frame. Although using faster scanners (resonant galvo with 12kHz scanning speed, or acoustic-optical deflector) could further improve the scanning speed, it is challenging to exceed 1k Hz frame rate. Both fibers are of 1km length. A flip mirror is inserted into the optical path after the DMD to split the input light field into SMF-based detection and MMF-based detection path.

Figure R2. Experimental setups for (a) MMF-based direct imaging and (b) MMF-based raster-scanning imaging system.

Representative recovered images by MMDN for 100s are presented in Figure R3. For slowly switching scenes, the 2D optical field for each frame is recorded by scanning the field along the x- and y-axis and measuring optical intensity by a photodetector, while the error rate of image recovery is about 5% due to the mechanical instability and system error. Most importantly, for fast scenes switching at 20kHz with SMF-based system, its slow imaging rate relative to the dynamic scene results in failed recovery of $\sim 50\%$ accuracy. To conclude, the MMF-based imaging technique gives rise to ultrafast imaging by one-shot without the requirement of slow and complex mechanisms.

Figure R3. Experimental results for (a) input patterns at representative timepoints using (b) MMF-based raster-scanning imaging system when the scene switches at 100Hz, (c) MMF-based raster-scanning imaging system when the scene switches at 20kHz, and (d) MMF-based direct imaging system when the scene switches at 20kHz.

- Second, regarding to the necessity of high-bandwidth optical communication, we experimentally demonstrated spatial transmission of over 1Gbit/s data rate basing on optical transmission module which has a bandwidth of 200MHz per channel. Specifically, we adopted a laser diode (VCSEL, 850nm) as the emitter and drove the on-off of the diode by a FPGA controlled circuit at 200MHz (see Fig. R4 (a)).

Figure R4. High-speed data optical communication experiments. (a) Optical setups and experimentally recovered signals of (b) single-channel transmission through SMF and (c) 5-channel transmission through MMF.

When using the standard SMF for signal transmission, one single channel of the diode is used, leading to a transmission data rate of 200 Mbit/s (see Fig. R4(b)). On the contrary, attributed to the spatial-multiplexing ability of MMF-based transmission system, we developed a laser array consisting of 5 segments that are independently programmable as the spatial-multiplexing transmission module at the input end. The spatial coupled output speckle pattern is detected by a PD array of 5×5-segment with 200MHz bandwidth for each detector. We experimentally demonstrated the decoupling of input signals at 200MHz per channel with the error rate less than 2.08×10^{-3} , revealing a total transmission data rate of 1.0 Gbit/s (see Fig. R4(c)). This experiment showcased convincing evidence of our proposed MMDN technique for spatial-multiplexing data transmission with enhanced capacity.

We have added the experimental comparisons between SMF and MMF-based technique for image transmission in the Discussion section and Supplementary Materials in the revised manuscript.

2. From response: “First, we would like to clarify that only in the pre-training stage do we need a large quantity of paired data, and the MMF is not required to maintain static during this procedure. This is because we use a neural network to model the transmission system, which offers higher robustness and generalization than transmission-matrix-based approaches on environmental instabilities [11].”

a. This is a good reference to compare. But the description is incorrect. On the contrary, reference 11 in response does not use transmission-matrix-based approaches. It actually uses very similar approaches to the presented work except it’s for dynamic diffuser rather than dynamic MMF.

b. How is this work compared to reference 11 in response and reference 18 in paper, in terms of innovation and performance?

Response: We thank the reviewer for the valuable comments.

a. We agree with the reviewer that reference 11 uses a similar neural network approach to achieve image recovery from scrambled measurement. Actually, it should be clarified that we cited this reference in the response to substantiate the assertion that “utilizing a neural network to model the transmission system provides higher robustness and generalization compared to transmission-matrix-based approaches on environmental instabilities”.

b. Reference 11 in response and reference 18 in paper exemplify two approaches to tackle descattering with adaptability to diverse scattering configurations. As outlined in the Introduction Section, ref 11 illustrates the utilization of diverse datasets (i.e., multiple sets of paired training data collected under different scattering conditions) to enhance the scalability of the trained model in unseen scattering conditions, while ref 18 showcases the application of a mixture of experts (MOE) framework that dynamically adapts to unseen scattering condition by adaptively weighting on several expert models.

- The primary drawback of the first category of solutions is that it sacrifices accuracy in exchange for generalized performance. For instance, as illustrated in Fig. 8 of ref 11, the Pearson correlation coefficient (PCC) for the recovered MNIST images through an unseen diffuser is only 0.626 for the best case. In contrast, our proposed

MMDN achieves >0.99 pixel-wise accuracy throughout the 1000s period (with the correlation of the fiber transmission model falling below 0.5). Besides, ref 11 necessitates the collection of training data under different scattering conditions, increasing the difficulty in practical implementation.

- The second category of solutions introduces an adjustable weight on several pre-trained models to achieve dynamic network synthesis. The adaptive mixture of multiple experts framework could capitalize on the high-accuracy property of each expert network and allow network adjustment to better suit present conditions in the meantime. However, given that only the weights to combine each expert are adjustable while the expert models remain fixed, its dynamic adjustability is quite confined. This limitation intuitively implies that the adaptability of the system is restricted to conditions not significantly distant from those where the expert networks were trained. From their experimental results of generalization to unseen scattering conditions in Fig. 4, we notice obvious performance deteriorations especially when the testing conditions remarkably deviate from the training conditions (for example, in Fig. 4(b), the Jaccard index drops from >0.9 to <0.75 as the diameter of scattering particles increases by 80%). For the long-term transmission task in our paper, it's scarcely feasible to build a look-up table to depict the fiber model variation states with only a few expert models.

In summary, attributed to its dynamic learning property, our MMDN outperforms previous fixed generalized models and partially adaptive network approaches in terms of accuracy and stability. This makes it particularly applicable for high-fidelity image recovery in scenarios involving dynamic scattering media with gradual changes.

References:

1. Ref 11 in response: Li Y., et al. Deep speckle correlation: a deep learning approach towards scalable imaging through scattering media, *Optica*, 2018.
2. Ref 18 in paper: Tahir W., et al. Adaptive 3D descattering with a dynamic synthesis network, *Light: Science & Applications*, 2022.

3. Rigor of paper writing, I didn't do a thorough check, but noticed that a few obvious citation errors. Please correct throughout. Here are the two instances that I noticed in main text:

- a. Reference 7 and 10 are the same one
- b. References 18 and 25 are the same one

Response: We have carefully read through the manuscript and corrected the citation errors.

4. As a minor point, I would suggest including Fig. R10 and R11 in the supplementary (surrounded by a proper discussion), which serve as a valuable reference for the audience who are interested in deploying the proposed method while the scarcity of training data is of concern.

Response: We thank the reviewer for the suggestion. In the revised version, we have included the Fig. R10 and R11 in the supplementary. In addition, a brief discussion on the requirement of large number of training sets is provided

in the Discussion Section “In contrast to conventional methods relying on the calibration of transmission matrices, deep learning provides an alternative pathway for addressing intricate reconstruction challenges, offering notable benefits such as enhanced accuracy and the avoidance of strict adherence to mathematical model priors. Nevertheless, the efficacy of deep learning is contingent upon a substantial collection of labeled samples for robust and generalized network training. We empirically examine the required size of the training dataset for MMDN, as illustrated in Supplementary Fig.11.”.

REVIEWERS' COMMENTS

Reviewer #3 (Remarks to the Author):

Review comments have been thoroughly addressed and I recommend acceptance as it is.